

# Argon offline-AMS source apportionment of organic aerosol over yearly cycles for an urban, rural and marine site in Northern Europe

C. Bozzetti[1], Y. Sosedova[1], M. Xiao[1], K. R. Daellenbach[1], V. Ulevicius[2], V. Dudoitis[2], G. Mordas[2], S. Byčenkienė[2], K. Plauškaitė[2], A. Vlachou[1], B. Golly[3], B. Chazeau[1], J.-L. Besombes[4], U. Baltensperger[1], J.-L. Jaffrezo[3], J. G. Slowik[1], El Haddad[1], I., and A. S. H. Prévôt[1]

[1] {Laboratory of Atmospheric Chemistry, Paul Scherrer Institute (PSI), 5232 Villigen-PSI, Switzerland}

[2] {Department of Environmental Research, SRI Center for Physical Sciences and Technology, LT-02300 Vilnius, Lithuania}

[3] {Université Grenoble Alpes, CNRS, LGGE, 38000 Grenoble, France}

[4] {Université Savoie Mont-Blanc, LCME, F-73000 Chambéry, France}

Correspondence to: A. S. H. Prévôt (andre.prevot@psi.ch); I. El Haddad (imad.el-haddad@psi.ch)

## Abstract

The widespread use of Aerodyne aerosol mass spectrometers (AMS) has greatly improved real-time organic aerosol (OA) monitoring, providing mass spectra that contain sufficient information for source apportionment. However, AMS field deployments remain expensive and demanding, limiting the acquisition of long-term datasets at many sampling sites. The offline application of aerosol mass spectrometry entailing the analysis of nebulized water extracted filter samples (offline-AMS) increases the spatial coverage accessible to AMS measurements, being filters routinely collected at many stations worldwide.

$PM_1$ (particulate matter with an aerodynamic diameter <1 µm) filter samples were collected during an entire year in Lithuania at three different locations representative of three typical





environments of the South-East Baltic region: Vilnius (urban background), Rūgšteliškis (rural
terrestrial), and Preila (rural coastal). Aqueous filter extracts were nebulized in Ar, yielding
the first AMS measurements of water-soluble atmospheric organic aerosol (WSOA) without
interference from air fragments. This enables direct measurement of the $CO^+$ fragment
contribution, whose intensity is typically assumed to be equal to that of $CO_2^+$. Offline-AMS
spectra reveal that the water soluble $CO_2^+$:$CO^+$ ratio not only shows values systematically <1
but is also dependent on season, with lower values in winter than in summer.
AMS WSOA spectra were analyzed using positive matrix factorization (PMF), yielding 5
factors: traffic exhaust OA (TEOA), biomass burning OA (BBOA), local OA (LOA)
contributing significantly only in Vilnius, and two oxygenated OA (OOA) factors
distinguished by seasonal variability. AMS-PMF source apportionment results were
consistent with those obtained from PMF applied to marker concentrations (i.e. major
inorganic ions, OC/EC, and organic markers including polycyclic aromatic hydrocarbons and
their derivatives, hopanes, long-chain alkanes, monosaccharides, anhydrous sugars, and lignin
fragmentation products). OA was the largest fraction of $PM_1$ and was dominated by BBOA
during winter with an average concentration of 2 μg m$^{-3}$ (53% of OA), while summer-OOA
(S-OOA), probably related to biogenic emissions was the prevalent OA source during
summer with an average concentration of 1.2 μg m$^{-3}$ (45% of OM).
PMF ascribed a large part of the $CO^+$ explained variability (97%) to the OOA and BBOA
factors. Accordingly we discuss a new $CO^+$ parameterization as a function of $CO_2^+$, and
$C_2H_4O_2^+$ fragments, which were selected to describe the variability of the OOA and BBOA
factors.
**1    Introduction**
Atmospheric aerosols affect climate (Lohmann et al., 2004, Schwarze et al., 2006), human
health (Dockery et al., 2005, Laden et al., 2000), and ecosystems on a global scale.
Quantification and characterization of the main aerosol sources are crucial for the
development of effective mitigation strategies. The Aerodyne aerosol mass spectrometer
(AMS, Canagaratna et al., 2007) and aerosol chemical speciation monitor (ACSM, Ng et al.,
2011, Fröhlich et al., 2013) have greatly improved air quality monitoring by providing real-
time measurements of the non-refractory (NR) submicron aerosol ($PM_1$) components.
Analysis of organic mass spectra using positive matrix factorization (PMF, Paatero, 1997;
Paatero and Tapper, 1994) has enabled the quantitative separation of OA factors, which can



be subsequently related to major aerosol sources and formation processes (e.g. Lanz et al.,
2007; Lanz et al., 2010; Zhang et al., 2011; Ulbrich et al., 2009; Elser et al., 2016 a). Despite
its numerous advantages, AMS field deployment remains expensive and demanding, and
therefore most of the studies are typically restricted to short-time periods and a single (or few)
sampling site(s). The limited amount of long-term datasets suitable for OA source
apportionment severely limits model testing and validation (Aksoyoglu et al., 2011;
Aksoyoglu et al., 2014; Baklanov et al., 2014), as well as for the development of appropriate
pollution mitigation strategies. AMS analysis of aerosol filter samples (Lee et al., 2011; Sun
et al., 2011; Mihara and Mochida, 2011; Daellenbach et al., 2016), which are routinely
collected at many stations worldwide, broadens the temporal and spatial scales available for
AMS measurements.
In this study we present the application of the offline-AMS methodology described by
Daellenbach et al. (2016) to yearly cycles of filter samples collected in parallel at three
different locations in Lithuania between September 2013 and August 2014. The methodology
consists of water extraction of filter samples, followed by nebulization of the liquid extracts,
and subsequent measurement of the generated aerosol by high-resolution time-of-flight AMS
(HR-ToF AMS). In this work, organic aerosol water extracts were nebulized in Ar, permitting
direct measurement of the $CO^+$ ion (Fig. S1), which is typically not directly quantified in
AMS data analysis due to interference with $N_2^+$, but is instead estimated as being equal to
$CO_2^+$ (Aiken et al., 2008). Direct measurement of $CO_2^+$ better captures the variability in the
total OA mass and its elemental composition as well as potentially improving source
apportionment of ambient aerosol. Aerosol elemental ratios and oxidation state are of
particular relevance as they provide important constraints for understanding aerosol sources,
processes, and for the development of predictive aerosol models (Canagaratna et al., 2015).
Aerosol composition in the south-east Baltic region has so far received little attention. To our
knowledge the only investigation of OA sources in this area was during a five-day period of
intense land clearing activity occurring in the neighboring Russian enclave of Kaliningrad
(Ulevicius et al., 2015; Dudoitis et al., 2016), in which transported biomass burning emissions
dominated the aerosol loading. OA source contributions under less extreme conditions remain
unstudied, with the most relevant measurements performed in Estonia with a mobile lab
during March 2014 at two different locations (Elser et al., 2016b). On-road measurements
revealed large traffic contributions with an increase of 20% from rural to urban environments.





Also, residential biomass burning (BB) and oxygenated OA (OOA) contributions were found
to be substantial.
In this study we present a complete source apportionment of the submicron OA fraction
following the methodology described by Daellenbach et al. (2016) in order to quantify and
characterize the main OA sources affecting the Lithuanian air quality. The three sampling
stations were situated in the Vilnius suburb (urban background), Preila (rural coastal
background), and Rūgšteliškis (rural terrestrial background), covering a wide geographical
domain and providing a good overview of the most typical Lithuanian and south-eastern
Baltic air quality conditions and environments. PMF analysis of offline-AMS measurements
are compared with the results reported by Ulevicius et al. (2015) and with PMF analysis of
chemical marker measurements obtained from the same filter samples.
**2  Sampling and offline measurements**
**2.1 Site description and sample collection**
We collected 24-h integrated $PM_1$ filter samples at 3 different stations in Lithuania from 30
September 2013 to 2 September 2014 using 3 High-Volume samplers (Digitel DHA80, and
DH-77) operating at 500 L $min^{-1}$. The particulate matter was collected on 150-mm diameter
quartz fiber filters (Pallflex Tissuquartz 2500QAT-UP / pure quartz, no binder) pre-baked at
800℃ for 8 h. Filter samples were wrapped in pre-baked aluminum foils (400℃ for 6 h),
sealed in polyethylene bags and stored at -20℃ after exposure. Field blanks were collected
and stored following the same procedure.
Sampling was conducted at urban (Vilnius), rural terrestrial (Rūgšteliškis) and rural coastal
(Preila) monitoring sites (Fig. 1). The rural terrestrial site of Rūgšteliškis serves as a baseline
against which urban-specific sources in the major population center of Vilnius can be
compared. The rural coastal site of Preila provides an opportunity to distinguish terrestrial and
marine sources.
The sampling station in Vilnius is located at the Center for Physical Sciences and Technology
campus (54°38' N, 25°10' E, 165 m a.s.l.) 12 km southwest of the city center (population:
535000) and is classified as an urban background site. The site is relatively far from busy
roads, and surrounded by forests to the north/northeast, and by a residential zone to the
south/east. It is ca. 350 km distant from the Baltic coast, and 98 km from the Rūgšteliškis
station (Fig. 1).





The station in Preila (55°55' N, 21°04' E, 5 m a.s.l.) is a representative rural coastal
background site, situated in the Curonian Spit National Park on the isthmus separating the
Baltic Sea from the Curonian Lagoon. The monitoring station is located <100 m from the
Baltic shore. The closest populated area is the village of Preila (population: 200 inhabitants),
located 2 km to the south.
The rural terrestrial station of Rūgšteliškis (55°26' N and 26°04' E, 170 m a.s.l.) is located in
the eastern part of Lithuania, about 350 km from the Baltic Sea. The site is surrounded by
forest and borders the Utenas Lake in the southwest. The nearest residential areas are
Tauragnai, Utena (12 km and 26 km west of the station, population: 32000 inhabitants) and
Ignalina (17 km southeast of the station, population: 6000 inhabitants).
**2.2 Offline-AMS analysis**
The term *offline-AMS* will be used herein to refer to the methodology described by
Daellenbach et al. (2016) and summarized below. For each analyzed filter sample, four 16-
mm diameter filter punches were subjected to ultrasonic extraction in 15 mL of ultrapure
water (18.2 M$\Omega$ cm at 25°C, total organic carbon (TOC) < 3 ppb) for 20 min at 30°C.
The choice of water instead of an organic solvent is motivated by two arguments:
-   Water yields the lowest background and hence the highest signal to noise compared to
other highly pure solvents (including methanol, dichloromethane and ethyl acetate).
-   In contrast to the water extraction, the use of organic solvents precludes the
quantification of the organic content in the extracts (e.g. by using a total OC analyzer),
which in turn prevents a quantitative source apportionment.
Liquid extracts were then filtered and atomized in Ar ($\geq$99,998 % Vol. abs., Carbagas, CH-
3073 Gümligen, Switzerland) using an Apex Q nebulizer (Elemental Scientific Inc., Omaha
NE 68131 USA) operating at 60°C. The resulting aerosol was then dried by passing through a
Nafion drier (Perma Pure, Toms River NJ 08755 USA), and subsequently analyzed by a HR-
ToF-AMS. 12 mass spectra per filter sample were collected (AMS V-mode, *m/z* 12-232, 30 s
collection time per spectrum). A measurement blank was recorded before and after each
sample by nebulizing ultrapure water for 12 minutes. Field blanks were measured following
the same extraction procedure as the collected filter samples, yielding a signal not statistically
different from that of nebulized milliQ water. Finally we registered the AMS fragmentation



spectrum of pure gaseous $CO_2$ (≥99,7 % Vol, Carbagas, CH-3073 Gümligen, Switzerland), in
order to derive its $CO_2^+$:$CO^+$ ratio.
Offline-AMS analysis was performed on 177 filter samples in order to determine the bulk
water-soluble organic matter (WSOM) mass spectral fingerprints. In total, 63 filters from
Rūgšteliškis, 42 from Vilnius, and 71 from Preila were measured in Ar. The reader is referred
to DeCarlo et al. (2006) for a thorough description of the AMS operating principles and
calibration procedures.
HR-ToF-AMS analysis software SQUIRREL (SeQUential Igor data RetRiEvaL, D. Sueper,
University of Colorado, Boulder, CO, USA) v.1.53G and PIKA (Peak Integration by Key
Analysis) v.1.11L for IGOR Pro software package (Wavemetrics, Inc., Portland, OR, USA)
were utilized to process and analyze the AMS data. HR analysis of the AMS mass spectra was
performed in the *m/z* range 12-115.

## 2.3 Supporting measurements

Additional offline analyses were carried out in order to validate and corroborate the offline-
AMS source apportionment results. This supporting dataset was also used as input for $PM_1$
source apportionment as discussed below. The complete list of the measurements performed
can be found in Table 1 and Table S1. Briefly, major ions were measured by ion
chromatography (IC; Jaffrezo et al., 1998); elemental and organic carbon (EC, OC) were
quantified by thermal optical transmittance following the EUSAAR2 protocol (Cavalli et al.,
2010); water-soluble OC (WSOC) was measured by water extraction followed by catalytic
oxidation and non-dispersive infrared detection of $CO_2$ using a total organic carbon analyzer
(Jaffrezo et al., 2005). Organic markers were determined by gas chromatography-mass
spectrometry (GC-MS; Golly et al., 2015); high performance liquid chromatography (HPLC)
associated with a fluorescence detector (LC 240 Perkin Elmer) and HPLC-pulsed
amperometric detection (PAD; Waked et al., 2014) for 67 composite samples. Composites
were created merging two consecutive filter samples, but no measurements are available for
Vilnius during summer. Measurements included 18 polycyclic aromatic hydrocarbons
(PAHs), alkanes (C21-C40), 10 hopanes, 13 methoxyphenols, 13 methyl-PAHs (Me-PAHs), 6
sulfur-containing-PAHs (S-PAHs), 3 monosaccharide anhydrides, and 4 monosaccharides
(including glucose, mannose, arabitol, and mannitol). In this work ion concentrations always
refer to the IC measurements.



1    Table 1. Overview of supporting measurements. A complete list of measured compounds can

2    be found in table S1.

| Analytical Method | Measured compounds | Filters measured |
|---|---|---|
| IC (Jaffrezo et al., 1998) | Ions | All |
| Thermal optical transmittance using Sunset Lab Analyzer (Birch and Cary, 1996) using EUSAAR2 protocol (Cavalli et al., 2010) | EC/OC | All |
| TOC analyzer using persulphate oxidation at 100°C of the OM, followed by $CO_2$ quantification with a non-dispersive infrared spectrophotometer (Jaffrezo et al., 1998) | WSOC | All |
| HPLC associated with fluorescence detector (LC 240 Perkin Elmer) (Golly et al., 2015, Besombes et al., 2001) | PAHs (table S1) | 67 composite samples |
| GC-MS (with and without derivatization step) (Golly et al., 2015) | S-PAHs, Me-PAHs, alkanes, hopanes, methoxyphenols, others | 67 composite samples |
| HPLC-PAD, (Waked et al., 2014) | Anhydrous sugars, sugars alcohols, monosaccharides | 67 composite samples |
| Chemiluminescence (Environnement S.A., Model AC31M) | $NO_x$ | Online (Vilnius only) |



In the following, subscripts *avg*, and *med* will denote average and median values,
respectively.

## 3 Source apportionment

Positive matrix factorization (PMF, Paatero and Tapper, 1994) is a bilinear statistical model
used to describe the variability of a multivariate dataset as the linear combination of a set of
constant factor profiles and their corresponding time series, as shown in Eq. (1):

$$x_{i,j} = \sum_{z=1}^{p}(g_{i,z} \cdot f_{z,j}) + e_{i,j} \qquad (1)$$

Here $x$, $g$, $f$, and $e$ denote elements of data, factor time series, factor profiles and residual
matrices, respectively, while subscripts $i,j$ and $z$ are indices for time, measured variables, and
factor number. The value $p$ represents the total number of factors chosen for the PMF
solution. The PMF algorithm iteratively solves Eq. (1) by minimizing the objective function
$Q$, defined in Eq. (2) Only non-negative $g_{i,z}$ and $f_{z,j}$ values are permitted:

$$Q = \sum_i \sum_j \left(\frac{e_{i,j}}{s_{i,j}}\right)^2 \qquad (2)$$

Here the $s_{i,j}$ elements represent entries in the input error matrix.
In this work the PMF algorithm was run in the robust mode in order to dynamically
downweigh the outliers. The PMF algorithm was solved using the multilinear engine-2 (ME-
2) solver (Paatero, 1999), which enables an efficient exploration of the solution space by *a*
*priori* constraining the $g_{i,z}$ or $f_{z,j}$ elements within a certain variability defined by the scalar *a*
($0 \leq a \leq 1$) such that the modelled $g_{i,z}$' and $f_{z,j}$' satisfy Eq. (3):

$$\frac{(1-a)f_{z,n}}{(1+a)f_{z,n}} \leq \frac{f_{z,n'}}{f_{z,m'}} \leq \frac{(1+a)f_{z,n}}{(1-a)f_{z,m}} \qquad (3)$$

Here $n$ and $m$ are any two arbitrary columns (variables) in the normalized F matrix. The
Source Finder toolkit (SoFi, Canonaco et al., 2013, v.4.9) for Igor Pro software package
(Wavemetrics, Inc., Portland, OR, USA) was used to configure the ME-2 model and for post-
analysis. PMF analysis was applied to two complementary datasets: (1) organic mass spectra
from offline-AMS measurements for the apportionment of OM sources and (2) molecular
markers for the apportionment of the measured $PM_1$ mass. These two analyses are discussed
separately below.



### 3.1 Offline-AMS PMF

In the following section we describe the offline-AMS source apportionment implementation, optimization and uncertainty assessment. Briefly, we selected the number of PMF factors based on residual analyses and solution interpretability; subsequently we explored the rotational uncertainty of our source apportionment model and discarded suboptimal solutions providing insufficient correlation of factor time series with external tracers. The offline-AMS source apportionment returns the water soluble PMF factor concentrations. Daellenbach et al. (2016) determined factor specific recoveries (including the extraction efficiencies), by comparing offline-AMS and online-ACSM source apportionments. Applying these recoveries enabled scaling the water soluble factor concentrations to the corresponding bulk OA concentrations. A sensitivity analysis of these recoveries was reported in Section 3.1.3, and the corresponding uncertainty was propagated to the source apportionment results.

A second selection step was carried out on the rescaled solutions as described in section 3.1.3. The offline-AMS source apportionment results presented in this study represent the average of the retained rescaled PMF solutions, while their variability represents our best estimate of the source apportionment uncertainty.

### 3.1.1 Inputs

The offline-AMS input matrices include in total 177 filter samples (62 filters from Rūgšteliškis, 42 from Vilnius, and 73 from Preila). Each filter sample was represented on average by 12 mass spectral repetitions to explore the effect of AMS and nebulizer stability on PMF outputs. A corresponding measurement blank was subtracted from each mass spectrum. The input PMF matrices included 269 organic fragments fitted in the mass range (12-115). The input error $s_{i,j}$ elements include the blank variability ($\sigma_{i,j}$) and the uncertainty related to ion counting statistic and ion-to-ion signal variability at the detector ($\delta_{i,j}$, Allan et al., 2003; Ulbrich et al., 2009):

$$s_{i,j} = \sqrt{\delta_{i,j}^2 + \sigma_{i,j}^2} \qquad (4)$$

We applied a minimum error to the $s_{i,j}$ matrix elements according to Ulbrich et al. (2009), and a down-weighting factor of 3 to all fragments with an average signal to noise lower than 2 (Ulbrich et al., 2009). Input data and error matrices were rescaled such that the sum of each row is equal to the estimated WSOM concentration, which is calculated as the product of the





measured WSOC multiplied by the OM:OC$_i$ ratios determined from the offline-AMS PMF
results.

### 3.1.2 Overview of retrieved factors and estimate of traffic exhaust OA (TEOA)

We used a 4-factor solution to represent the variability of the input data. The 4 separated OA
factors included the following:
1/ a biomass burning OA (BBOA) factor highly correlated with levoglucosan originating from
cellulose pyrolysis;
2/ a local OA (LOA) factor explaining a large fraction of N-containing fragments variability
and contributing mostly in Vilnius during summer and spring;
3/ a background oxygenated-OA (B-OOA) factor showing relatively stable contributions at all
seasons;
4/ a summer-OOA (S-OOA) factor showing increasing concentrations with the average daily
temperature.
If the number of factors is decreased to 3, a mixed BBOA/B-OOA factor is retrieved, and
significant structure appears in the residuals during winter (Fig. S2, S3, S4). Increasing the
number of factors to 5 and 6, leads to a splitting of OOA factors that cannot be interpreted in
terms of specific aerosol sources/processes (Fig. S2, S3). The further separated OOA factor in
the 5-factor solution possibly derived from the splitting of B-OOA; in fact the sum of the
newly separated OOA and B-OOA in the 5-factor solution correlated well with the B-OOA
time series from the 4-factor solution ($R = 0.93$). Overall, a clear structure removal in the
residual time-series was observed until a number of factors equal to 4 (Fig. S4, S5).
We also explored a 5-factor solution in which a hydrocarbon-like OA (HOA) profile from
Mohr et al. (2012) to estimate the TEOA contribution. However, the water-soluble TEOA
(WSTEOA) contribution to WSOM was estimated as 0.2%$_{avg}$ (section 3.1.4), likely too small
for PMF to resolve. We performed 100 PMF runs by randomly varying the HOA $a$-value. The
obtained results showed a low TEOA correlation with hopanes ($R_{max} = 0.25$, $R_{min} = -0.15$)
with 45% of the PMF runs associated with negative Pearson correlation coefficients,
supporting the hypothesis that this factor has too small a contribution to be resolved.
Therefore, we selected the 4-factor solution as our best representation of the data, while





TEOA was instead estimated by a chemical mass balance (CMB) approach and not based on
AMS mass spectral features.
TEOA concentrations are estimated using a CMB approach that assumes hopanes, present in
lubricant oils engines, (Subramanian et al., 2006) to be unique tracers for traffic. However,
hopanes can also be emitted upon combustion of different types of fossil fuel, in particular by
coal combustion (Rutter et al., 2009), therefore the traffic contribution estimated here,
although very small (as discussed in the result section) should be considered as an upper
estimate. Still, the EC/hopanes ratio determined in this work (900±100) is consistent with
EC/hopanes for TE (1400±900: He et al., 2006; He et al., 2008; El Haddad et al., 2009; Fraser
et al., 1998) and not with the coal EC/hopanes from literature profiles (300±200: Huang et al.,
2014; supplementary information (SI)). To assess the traffic exhaust OC (TEOC) contribution
we used the sum of the four most abundant hopanes (17a(H),21b(H)-norhopane,
17a(H),21b(H)-hopane,     22S,17a(H),21b(H)-homohopane,     and     22R,17a(H),21b(H)-
homohopane (hopanes$_{sum}$)). The TEOC contribution was estimated from the average
hopanes$_{sum}$/TEOC ratio (0.0012±0.0005) from tunnel measurements reported by He et al.
(2006), He et al. (2008), El Haddad et al. (2009), and Fraser et al. (1998), where the four
aforementioned hopanes were also the most abundant. In order to rescale TEOC to the total
TEOA concentration we assumed an OM:OC$_{TEOA}$ ratio of 1.2±0.1 (Aiken et al., 2008, Mohr
et al., 2012, Docherty et al., 2011, Setyan et al., 2012). The uncertainty of the estimated
TEOA concentration was assessed by propagating the uncertainties relative to the
OM:OC$_{TEOA}$ ratio (8.3%), the hopanes$_{sum}$/TEOC ratio (41.7%), the hopane measurement
repeatability (11.5%), and detection limits (7 pg m$^{-3}$).

### 3.1.3. Source apportionment uncertainty

A common issue in PMF is the exploration of the rotational ambiguity, here addressed by
performing 100 PMF runs initiated using different input matrices. We adopted a bootstrap
approach (Davison and Hinkley, 1997) to generate the new input data and error matrices
(Brown et al., 2015). Briefly, the bootstrap algorithm generates new input matrices by
randomly resampling mass spectra from the original input matrices. As already mentioned,
the input matrices contained ca. 12 mass spectral repetitions per filter sample; therefore the
bootstrap approach was implemented in order to resample random filter sample mass spectra
together with the corresponding measurement repetitions. Each newly generated PMF input



matrix had a total number of samples equal to the original matrices (177 samples), although
some of the original 177 filter samples are represented several times, while others are not
represented at all. Overall we resampled on average 63±2% of the filter samples per bootstrap
run. The generated data matrices were finally perturbed by varying each $x_{i,j}$ element within
twice the corresponding uncertainty ($s_{i,j}$) assuming a normal distribution of the errors.
Solutions were selected and retained according to three acceptance criteria. Solutions were
selected and retained according to three acceptance criteria based on PMF factor correlations
with corresponding tracers: BBOA vs. levoglucosan, B-OOA vs. $NH_4^+$, and S-OOA vs.
average daily temperature. In order to discard suboptimal PMF runs, we only retained
solutions associated with positive Pearson correlation coefficients for each criterion, for both
the individual stations and the entire dataset. In total 95% of the solutions were retained
following this approach.
The offline-AMS PMF analysis provides the water-soluble contribution of the identified
aerosol sources. In order to rescale the water-soluble organic carbon concentration of a
generic factor $z$ (WSZOC) to its total OC concentration (ZOC) we used the factor recoveries
($R_Z$) determined by Daellenbach et al. (2016) according to Eq. (5):

$$ZOC_i = \frac{WSZOC_i}{R_z} \tag{5}$$

For each PMF factor (BBOA, W-OOA, and S-OOA), the water-soluble organic carbon
contribution was determined from the OM:OC ratio calculated from the (water-soluble) factor
mass spectrum (Aiken et al. 2008). For LOA, whose recovery was not previously reported,
$R_{LOA}$ was estimated from a single parameter fit according to Eq. (6)

$$OC = TEOC + \frac{WSBBOA}{(OM/OC)_{WSBBOA} \cdot R_{BBOA}} + \frac{WSW-OOA}{(OM/OC)_{WSS-OOA} \cdot R_{OOA}} + \frac{WSS-OOA}{(OM/OC)_{WSB-OOA} \cdot R_{OOA}} + \frac{WSLOA}{(OM/OC)_{LOA} \cdot R_{LOA}} \tag{6}$$

For each of the 95 retained PMF solutions, Eq. (6) was fitted 100 times by randomly selecting
a set of 100 $R_{BBOA}$, $R_{OOA}$ value combinations from those determined by Daellenbach et al.
(2016). Each fit was initiated by perturbing the input $OC_i$ and $TEOC_i$ within their
uncertainties, assuming a normal distribution of the errors. In order to explore the effect of
possible bulk extraction efficiency (WSOC/OC) systematic measurement biases on our $R_Z$
estimates, we also perturbed the OC, WSOC, $R_{BBOA}$, and $R_{OOA}$ (Daellenbach et al., 2016)
inputs. Specifically, we assumed an estimated accuracy bias of 5% for each of the perturbed
parameters, which corresponds to the OC and WSOC measurement accuracy. In total $9.5 \cdot 10^3$
fits were performed (Eq. 6) and we retained only solutions with average OC residuals not



statistically different from 0 within 1σ for each station individually and for summer and
winter individually (~8% of the $9.5 \cdot 10^3$ fits, Fig. S6). The OC residuals of the accepted
solutions did not manifest a clear correlation with the LOA concentration (Fig. S7), indicating
that the estimated $R_{LOA}$ was properly fitted, without compensating for unexplained variability
of the PMF model or biases from the other $R_z$. $R_z$ distributions shown in Fig. S8 accounted for
all uncertainties and biases mentioned above. $R_{LOA,med}$ was estimated to be equal to 0.66 (1st
quartile 0.61, 3rd quartile 0.69, Fig. S8), while the retained $R_{BBOA}$ and $R_{OOA}$ values ($R_{BBOA,med}$
0.57, 1st quartile 0.55, 3rd quartile 0.60; $R_{OOA,med}$ 0.84, 1st quartile 0.81, 3rd quartile 0.88) were
systematically lower than those reported by Daellenbach et al. (2016), reflecting the lower
bulk extraction efficiency (bulk EE = WSOC/OC) measured for this dataset (median = 0.59,
1st quartile = 0.51, 3rd quartile = 0.72 *vs.* median = 0.74, 1st quartile = 0.66, 3rd quartile 0.90 in
Daellenbach et al. (2016)). All the retained $R_k$ combinations are available at DOI:
doi.org/10.5905/ethz-1007-53.
Source apportionment uncertainties ($\sigma_{S.A.}$) were estimated for each sample *i* and factor *z* as the
standard deviation of all the retained PMF solutions (~8% of the $9.5 \cdot 10^3$ fits). In addition to
the rotational ambiguity of the PMF model (explored by the bootstrap technique) and $R_Z$
uncertainty, each PMF solution included on average 10 repetitions for each filter sample, and
hence $\sigma_{S.A.}$ accounted also for measurement repeatability. In this work, the statistical
significance of a factor contribution is calculated based on $\sigma_{S.A.,z,i}$ (Tables S2 and S3).
### 3.1.4. Sensitivity of PMF to the un-apportioned TEOA fraction
Despite representing only a small fraction, the un-apportioned water-soluble TEOA
(WSTEOA) contribution could in theory affect the apportionment of the other sources in the
PMF model. To assess this, we performed a PMF sensitivity analysis by subtracting the
estimated WSTEOA concentration from the input PMF data matrix, and by propagating the
estimated WSTEOA uncertainty (section 3.1.2) in the input error matrices. To estimate the
WSTEOA concentration we assumed $R_{TEOA}$ of 0.11±0.01 (Daellenbach et al., 2016) and we
used the HOA profile reported by Mohr et al. (2012) as surrogate for the TEOA mass spectral
fingerprint. This approach is equivalent to constraining both the WSTEOA time series and
factor profile. Overall the WSTEOA contribution to WSOM was estimated as $0.2\%_{avg}$,
making a successful retrieval of WSTEOA unlikely (Ulbrich et al., 2009). Consistently, PMF
results obtained from this sensitivity analysis indicated that BBOA and B-OOA were robust,
showing only 1% difference from the average offline-AMS source apportionment results,





with BBOA increased and B-OOA decreased. S-OOA and LOA instead showed larger
deviations from the average source apportionment results (S-OOA increased by 8% and LOA
decreased by 15%), yet within our source apportionment uncertainties. These results highlight
the marginal influence of the un-apportioned WSTEOA fraction on the other factors.
## 3.2 Marker-PMF: measured $PM_1$ source apportionment
In the following section we describe the implementation of source apportionment using
chemical markers (marker-PMF), as well as its optimization and uncertainty assessment. We
discuss the number of factors and the selection of specific constraints to improve the source
separation. Subsequently we discuss the source apportionment rotational uncertainty, and the
sensitivity of our PMF results to the number of source specific markers, and to the assumed
constraints.
### 3.2.1 Inputs
The marker-PMF yields a source apportionment of the entire measured $PM_1$ fraction (organic
and inorganic). Measured $PM_1$ is defined here as the sum of EC, ions measured via IC, and
OM estimated from OC measurements multiplied by the $(OM:OC)_i$ ratio determined from the
offline-AMS PMF results by summing the factor profiles OM:OC ratios weighted by the time
dependent factor relative contributions (rescaled by the recoveries). PMF was used to analyze
a data matrix consisting of selected organic molecular markers, ions measured by IC, EC, and
the remaining OM fraction ($OM_{res}$) calculated as the difference between OM and the sum of
the organic markers already included in the input matrix. The marker-PMF analysis is limited
by the lack of elemental measurements (e.g. metals and other trace elements) typically used to
identify mineral dust and certain anthropogenic sources. All markers showing concentrations
above the detection limits for more than 25% of the samples were selected as input variables
(72 in total). The PMF input matrices contain 67 composite samples (31 for Rūgšteliškis, 29
for Preila, and 7 for Vilnius). The errors ($s_{i,j}$) were estimated by propagating for each $j$
variable the detection limits (DL) and the relative repeatability ($RR$) multiplied by the $x_{i,j}$
concentration according to Eq. (7) (Rocke and Lorenzato, 1995):

$$s_{i,j} = \sqrt{(DL_j{}^2 + (x_{i,j} \cdot RR_{i,j})^2)} \tag{7}$$



### 3.2.2 Number of factors and constraints

We selected a 7-factor solution to explain the variability of the measured $PM_1$ components. The retrieved factors were biomass burning (BB), traffic exhaust (TE), primary biological organic aerosol (PBOA), $SO_4^{2-}$-related secondary aerosol (SA), $NO_3^-$-related SA, methane sulfonic acid (MSA)-related SA, and a $Na^+$-rich factor explaining the variability of inorganic components typically related to resuspension of mineral dust, sea salt, and road salt.

We first tested an unconstrained source apportionment. This led to a suboptimal separation of the aerosol sources, with large mixings of PMF factors associated with contributions of markers originating from different sources. In particular we observed mixing of BB markers (e.g. levoglucosan) with fossil fuel combustion markers such as hopanes, as well as with inorganic ions such as $NO_3^-$ and $Ca^{2+}$. All these markers, although related to different emission/formation processes, are characterized by similar seasonal trends, i.e. higher concentrations during winter than in summer. Specifically, the BB tracers increase during winter because of domestic heating activity, hopanes presumably because of the accumulation in a shallower boundary layer and lower photochemical degradation, $NO_3^-$ because of the partitioning into the particle phase at low temperatures, and $Ca^{2+}$ because winter was the windiest season and therefore was associated with the most intense resuspension.

We subsequently exploited the markers' source-specificity to set constraints for the profiles output by our model: for each individual source, we treated the contribution of the unrelated source-specific markers as negligible (e.g. we assumed that TE, SA, Na-rich factor and PBOA do not contribute to levoglucosan). In contrast, the non-source specific variables were freely apportioned by the PMF algorithm. In a similar way we set constraints for primary markers (e.g. $K^+$ and $Ca^{2+}$) and combustion related markers (e.g. PAHs), which are not source-specific but the contribution of which can be considered as negligible in the SA factors. In this case the algorithm can freely apportion these markers to all the primary factors and combustion-related factors, respectively.

In details, EC, PAHs, and methyl-PAHs were constrained to zero in non-combustion sources, i.e. all profiles but TE and BB. While EC could partially derive from dust resuspension, literature profiles for this source suggest an EC contribution below 1% (Chow et al., 2003). This is expected to be also the case here given the distance of the three stations from residential areas and busy roads. Methoxyphenols and sugar anhydrides, considered to be unique BB markers, were constrained to zero in all sources but BB. Similarly, hopanes were





constrained to zero in all factors but TE. We also assumed no contribution from glucose,
arabitol, mannitol, and sorbitol to all secondary factors, and traffic exhaust. The $SO_4^{2-}$
contribution from primary traffic emissions was estimated to be negligible, given the use of
desulfurized fuel for vehicles in Lithuania. Likewise, alkane contributions were assumed to be
zero in the SA factors, similar to the contribution of $Ca^{2+}$, $Na^+$, $K^+$ and $Mg^{2+}$ in the SA factors
and TE.
The number of factors was increased until no mixing between source-specific markers for
different aerosol sources/processes was observed any more. Secondary sources instead were
explained by three factors because of the distinct seasonal and site-to-site variability of MSA,
$NO_3^-$ and $SO_4^{2-}$. Oxalate correlated well with $NH_4^+$ ($R=0.62$) and the latter well with the sum
of $SO_4^{2-}$ and $NO_3^-$ equivalents ($R=0.98$). Note that the aforementioned secondary tracers were
not constrained in any factor with the exception of $SO_4^{2-}$ contributions which were assumed to
be negligible in the TE factor. Moreover the 7-factor solution showed unbiased residuals
(residual distribution centered at 0 within $1\sigma$) for all the stations together and for each station
individually, while lower order solutions showed biased residuals for at least one station or all
the stations together.
PMF results obtained assuming only the aforementioned constraints returned suboptimal
apportionments of $OM_{res}$ and $Na^+$ between the BB and the $Na^+$-rich factor, with unusually
high $OM_{res}$ fractional contributions in the $Na^+$-rich factor and unusually high $Na^+$
contributions in the BB profile in comparison with literature profiles (Chow et al., 2003;
Huang et al., 2014 and references therein; Schauer et al., 2001). Similarly the $EC/OM_{res}$ value
for TE was substantially lower than literature profiles (El Haddad et al., 2013 and references
therein). Other constraints were therefore introduced to improve the separation of these three
variables. Specifically, EC and $OM_{res}$ were constrained in the traffic profile to be equal to
0.45 and 0.27 ($a$-value = 0.5) according to El Haddad et al. (2013), while EC was constrained
to 0.1 ($a$-value = 1) in the BB profile according to Huang et al. (2014) and references therein.
$Na^+$ was constrained to 0.2% ($a$-value = 1) in BB according to Schauer et al. (2001), while
$OM_{res}$ was constrained to zero in the $Na^+$-rich factor to avoid mixing with BB. Although this
represents a strict constraint, we preferred avoiding constraining $OM_{res}$ to a specific value for
the $Na^+$-rich factor which could not be linked to a unique source but possibly represents
different resuspension-related sources (e.g. sea salt, mineral dust and road dust). However, we
expect none of the aforementioned sources to explain a large fraction of the submicron $OM_{res}$




(the OC:dust ratio for dust profiles is 1-15% according to Chow et al., 2003). The sensitivity
of our source apportionment to the constraints listed in this section is discussed in the next
section.
### 3.2.3. Source apportionment uncertainty and sensitivity analyses
We explored the model rotational uncertainty by performing 20 bootstrap PMF runs, and by
perturbing each input $x_{i,j}$ element within $2 \cdot s_{i,j}$ assuming a normal distribution of the errors.
Results and uncertainties of the PMF model reported in this paper represent the average and
the standard deviation of the bootstrap runs.
We tested the sensitivity of our solution to the constraints listed in section 3.2.2. All the
constraints assuming variable contributions equal to zero were loosened, assuming for each
variable a contribution equivalent to 50%, 37.5%, 25%, and 12.5% of its average relative
contribution to measured $PM_1$. In all cases the *a*-value was set to 1. As expected, results
showed better agreement with the fully constrained solution in the cases of stronger
constraints, meaning that the highest agreements were observed for the 12.5% case both in
terms of mass balance and factor time-series correlations (Fig. S9). The average factor
concentrations for the 12.5% case and the fully constrained average bootstrap PMF solutions
were not statistically different (confidence interval of 95%). Statistically significant
differences arose for the of the $SO_4^{2-}$-related SA in the 50% and 37.5% cases, and the $Na^+$-
rich factor in the 25% and 37.5% cases, indicating that loosening the constraints allowed
additional rotational uncertainty in comparison to the uncertainty explored by the bootstrap
approach. By contrast, the factors associated with large relative uncertainties from the marker
source apportionment (TE and PBOA, Table S3) showed the best agreement in terms of
concentrations (Fig. S9) with the fully constrained solution, suggesting that the variability
introduced by loosening the constraints did not exceed that already accounted for by the
bootstrap approach. As previously mentioned, the largest contribution discrepancies were
observed for the $SO_4^{2-}$-related SA and $Na^+$-rich factor. Looser constraints increased the
explained variability of primary components such as EC, arabitol, sorbitol, $K^+$, $Mg^{2+}$, and
$Ca^{2+}$ by the (secondary) $SO_4^{2-}$-related SA factor. The $Na^+$-rich factor showed increasing
contributions from $OM_{res}$ and from BB components such as methoxyphenols, and anhydrous
sugars, which exhibited similar seasonal trends as the $Na^+$-rich factor. None of the marker-
PMF factors showed statistically different average contributions (confidence interval of 95%)



when tolerating a variability of the constrained variables within 12.5% of their relative
contribution to $PM_1$. Note that with this degree of tolerance the contribution of OM to the
$Na^+$-rich was 28%, which is unrealistically high compared to typically reported values for
OM:dust ratios (<15% Chow et al., 2003). Therefore, we consider the fully constrained PMF
solution to represent best the average composition of the contributing sources.
The marker-PMF source apportionment depends strongly on the input variables (i.e. measured
markers), as these are assumed to be highly source specific. That is, minor sources, such as
MSA-related SA and PBOA, are separated because source-specific markers were used as
model inputs. Meanwhile, more variables were used as tracers for TE and BB
(methoxyphenols (5 variables), sugar anhydrides (3 variables), and hopanes (5 variables)),
which gives more weight to these specific sources. We explored the sensitivity of the PMF
results to the number and the choice of traffic and wood burning markers, by replacing them
with randomly selected input variables. In total 20 runs were performed and the average
contribution of the different sources to $OM_{res}$ was compared with the marker source
apportionment average results, where bootstrap was applied to resample time points. Results
displayed in Fig. S10 are in agreement the apportionment of $OM_{res}$ from BB within $11\%_{avg}$,
highlighting its robustness. The agreement for TE was lower, which is not surprising given
the lower contribution of this source and the smaller number of specific markers (hopanes).
However, these uncertainties were within the marker source apportionment uncertainty (Fig.
S10), implying that the results were not significantly sensitive to the number and the choice of
input markers for BB and traffic exhaust.

## 23   4   Results and Discussion

### 24   4.1 $PM_1$ composition

An overview of the measured $PM_1$ composition can be found in Fig. 1. Measured $PM_1$
average concentrations were in general low, with lower values detected at the rural terrestrial
site of Rūgšteliškis ($5.4\ \mu g\ m^{-3}_{avg}$) than in Vilnius ($6.7\ \mu g\ m^{-3}_{avg}$) and Preila ($7.0\ \mu g\ m^{-3}_{avg}$).
OM represented the major fraction of measured $PM_1$ for all seasons and stations, with $57\%_{avg}$
of the mass. The average OM concentrations were higher during winter ($4.2\ \mu g\ m^{-3}$) than in
summer ($3.0\ \mu g\ m^{-3}$) at all sites probably to a combination of domestic wood burning activity
and accumulation of the emissions in a shallower boundary layer. For similar reasons, EC



1. average concentrations showed higher values during winter (0.42 µg m$^{-3}$) than in summer

2. (0.25 µg m$^{-3}$). During summer, the average EC concentration was ∼5 times higher in Vilnius

3. (0.54 µg m$^{-3}$) than in Preila and Rūgšteliškis (0.12 and 0.11 µg m$^{-3}$, respectively), indicating

4. an enhanced contribution from combustion emissions. In the absence of domestic heating

5. during this period, a great part of these emissions may be related to traffic. During winter, EC

6. concentrations were comparable at all sites (only 25% higher in Vilnius than in Preila and

7. Rūgšteliškis). This suggests that a great share of wintertime EC may be related to BB, the

8. average contribution of which is significant at all stations within 3σ (table S2). It should be

9. noted that the highest measured PM$_1$ concentrations were detected at the remote rural coastal

10. site of Preila during three different pollution episodes. In particular, the early March episode

11. corresponded to the period analyzed by Ulevicius et al. (2015) and Dudoitis et al. (2015), and

12. was attributed to regional transport of polluted air masses associated to an intense land

13. clearing activity characterized by large scale grass burning in the neighboring Kaliningrad

14. region. SO$_4^{2-}$ represented the second major component of measured PM$_1$ (20%$_{med}$) at all sites

15. and seasons. Its average concentration remained rather constant with only slightly higher

16. concentrations in summer than in winter (1.2±0.7 µg m$^{-3}$, and 1.1±0.6 µg m$^{-3}$ respectively).

17. Overall SO$_4^{2-}$ concentrations did not show large differences from site-to-site, suggestive of

18. regional sources. By contrast NO$_3^-$ showed a clear seasonality with larger contributions in

19. winter (average 0.9±0.8 µg m$^{-3}$ equivalent to 12% of measured PM$_1$) than in summer

20. (0.03±0.03 µg m$^{-3}$), as expected from its semi-volatile nature.

21. **4.2 OM source apportionment (Offline-AMS PMF)**

22. The apportioned PMF factors were associated to aerosol sources/processes according to their

23. mass spectral features, seasonal contributions and correlations with tracers. The four

24. identified factors were BBOA, LOA, B-OOA, and S-OOA, which are thoroughly discussed

25. below. The TEOA contributions instead were determined using a CMB approach.

26. BBOA was identified by its mass spectral features, with high contributions of C$_2$H$_4$O$_2^+$, and

27. C$_3$H$_5$O$_2^+$ (Fig. 2), typically associated with levoglucosan fragmentation from cellulose

28. pyrolysis (Alfarra et al., 2007), accordingly the BBOA factor time series correlated well with

29. levoglucosan (Pearson correlation coefficient: $R$=0.90, Fig. S11). BBOA contributions were

30. higher during winter and lower during summer (Fig. 3a). We determined the biomass burning

31. organic carbon (BBOC) concentration from the BBOA time series divided by the





OM:OC$_{BBOA}$ ratio determined from the corresponding HR spectrum. The winter
levoglucosan/BBOC ratio was 0.16$_{med}$, consistent with values reported in continental Europe
for ambient BBOC profiles (Zotter et al., 2014; Minguillón et al., 2011; Herich et al., 2014).
The second factor was defined as LOA because of its statistically significant contribution
(within 3σ) only in Vilnius during summer (table S2), in contrast to other potentially local
primary (e.g. BBOA) and secondary (S-OOA) sources which contributed at all sites. The
LOA mass spectrum was characterized by a high contribution of N-containing fragments
(especially $C_5H_{12}N^+$, and $C_3H_8N^+$), with the highest N:C ratio (0.049) among the apportioned
PMF factors (0.029 for BBOA, 0.013 for S-OOA, 0.023 for B-OOA). A similar factor was
also observed by Byčenkienė et al. (2016) using an ACSM at the same station. In that work,
high LOA concentrations were associated with wind directions from N-NW, and the authors
suggested the sludge utilization system of Vilnius (UAB Vilniausvandenys) situated 3.9 km
NW from the sampling station as a probable source.
Two different OOA sources (S-OOA and B-OOA) were resolved and exhibited different
seasonal trends. Separation and classification of OOA sources from offline-AMS is typically
different from that of online AMS and ACSM measurements, mainly due to the different time
resolution. Online-AMS OOA factors are commonly classified based on their volatility (semi-
volatile OOA and low-volatility OOA). This differentiation is typically achieved only for
summer datasets when the temperature gradient between day and night is sufficiently high,
yielding a detectable daily partitioning cycle of the semi-volatile organic compounds and
$NO_3^-$ between the gas and the particle phases. Online AMS datasets have higher time
resolution than filter sampling, but sampling periods typically cover only a few weeks.
Therefore the apportionment is driven by daily variability rather than seasonal differences. By
contrast, in the offline-AMS source apportionment, given the 24-h time resolution of the filter
sampling and the yearly cycle time coverage, the separation of the factors is driven by the
seasonal variability of the sources and by the site-to-site differences. Therefore, the offline-
AMS source apportionment separates factors by seasonal trends rather than volatility.
The resolved B-OOA factor explained a higher fraction than S-OOA. It was associated with
background oxygenated aerosols as no systematic seasonal pattern was observed. However,
B-OOA correlated well with $NH_4^+$ ($R$=0.69, Fig. S11), and had the highest OM:OC ratio
among the apportioned PMF factors (2.21).




Unlike B-OOA, S-OOA showed a clear seasonality with higher contributions during summer,
increasing exponentially with the average daily temperature (Fig. S12a). During summer the
site-to-site S-OOA concentrations were not statistically different within a confidence interval
of 95%, while during winter the site-to-site agreement was lower, possibly due to the larger
model uncertainty associated with the low S-OOA concentrations. A similar S-OOA *vs.*
temperature relationship was reported by Leaitch et al. (2011) for a terpene dominated
Canadian forest using an ACSM and by Daellenbach et al. (2016) and Bozzetti et al. (2016)
for the case of Switzerland (Fig. S12b), using a similar source apportionment model. This
increase in S-OOA concentration with temperature is consistent with the exponential increase
in biogenic SOA precursors (Guenther et al., 2006). Therefore, even though the behavior of S-
OOA at different sites might be driven by several parameters, including vegetation coverage,
available OA mass, air masses photochemical age and ambient oxidation conditions (e.g. $NO_x$
concentration), temperature seems to be the main driver of S-OOA concentrations. Overall
more field observations at other European locations are needed to validate this relation. While
the results indicate a probable secondary biogenic origin of the S-OOA factor, the precursors
of the B-OOA factor are not identified. In section 4.4.2 more insights into the OOA sources
will be discussed.
The S-OOA profile showed a $CO_2^+/C_2H_3O^+$ ratio of $0.61_{avg}$, placing it in the region of semi-
volatile SOA from biogenic emissions in the $f44/f43$ space (Ng et al., 2011), as attributed by
Canonaco et al. (2015). Despite the higher summer photochemical activity, the water-soluble
bulk OA showed more oxidized mass spectral fingerprints during winter ($O:C=0.61_{avg}$) than
in summer ($O:C=0.55_{avg}$), similar to the results presented by Canonaco et al. (2015) for
Zurich. Accordingly, the S-OOA profile also showed a less oxidized water-soluble mass
spectral fingerprint than B-OOA, with an O:C ratio of $0.40_{avg}$, in comparison with $0.80_{avg}$ for
B-OOA. Considering the sum of B-OOA and S-OOA, the median $OOA:NH_4^+$ ratios for
Rūgšteliškis, Preila, and Vilnius were 3.2, 2.4, and 2.5 respectively, higher than the average
but within the range of the values reported by Crippa et al. (2014) for 25 different European
rural sites ($2.0_{avg}$; minimum value 0.3; maximum 7.3).
**4.3 PM$_1$ source apportionment (marker-PMF)**
The PMF factors in this analysis were associated with specific aerosol sources/processes
according to their profiles, seasonal trends and relative contributions to the key variables. Fig.



4 displays factor profiles, and the relative contribution of each factor to each variable. The
$Na^+$-rich factor explained a large part of the variability of $Ca^{2+}$, $Mg^{2+}$, and $Na^+$ (Fig. 4) and
showed higher contributions during winter than in summer, suggesting a possible
resuspension of sand and salt typically used during winter in Lithuania for road de-icing. This
seasonal trend is also consistent with wind speed, which showed the highest monthly values
during December 2013 and January 2014. We cannot exclude the possibility that this factor
may include contributions from sea salt, although $Na^+$ and $Cl^-$ were not enhanced at the
marine station in comparison with the other stations. The overall contribution of this $Na^+$-rich
factor to measured $PM_1$ was relatively small ($1\%_{avg}$), but may be larger in the coarse fraction.
The BB factor showed a well-defined seasonality, with high contributions during winter. This
factor explained a large part of the variability of typical wood combustion tracers such as
methoxyphenols, sugar anhydrides (including levoglucosan, mannosan, and galactosan), $K^+$,
$Cl^-$, EC, PAHs, and methyl-PAHs (Fig. 4). Using the ratio (1.88) calculated from offline-
AMS, we estimated the levoglucosan:BBOC ratio to be $0.18_{avg}$, which is within the range of
previous studies (Ulevicius et al., 2015 and references therein). Note that this factor explained
also large fractions of variables typically associated with non-vehicular fossil fuel
combustion, such as benzo(b)naphtho(2,1-d)thiophene (BNT[2,1]) and 6,10,14-trimethyl-2-
pentadecanone (DMPT, Fig. 4, Manish et al., 2007; Subramanian et al., 2007), indicating a
potential mixing of BB with fossil fuel combustion sources. However, the fossil fuel
combustion contribution to BB is unlikely to be large, considering the low concentrations of
fossil fuel tracers such as hopanes (66% of the samples below quantification limit (<QL)),
BNT[2,1] (64%<QL), and DMPT (55%<QL). Moreover, the above mentioned agreement of
the levoglucosan:BBOC ratio with previous studies corroborates the BB estimate from the
marker-PMF.
The traffic exhaust factor explained a significant fraction of the alkane variability, with a
preferential contribution from light alkanes (Fig. 4). Its contribution was statistically
significant within $1\sigma$ only for one filter collected in Vilnius. However on average the
concentration was higher in Vilnius than at the other stations and in general higher in winter
than in summer.
The PBOA factor explained the variability of the primary biological components, such as
glucose, mannitol, sorbitol, arabitol, and alkanes with an odd number of carbon atoms
(consistent with Bozzetti et al., 2016 and references therein). Highest PBOA concentrations



were observed during spring, especially at the rural site of Rūgšteliškis. Overall the
contribution of this factor was uncertain with an average relative model error of 160%
probably due to the small PBOA contributions ($0.6\%_{avg}$ of the total OM), which hampers a
more precise determination by the model. In particular $OM_{res}$ was the variable showing the
highest mass contribution to the PBOA factor. However, the large contribution and the large
uncertainty of $OM_{res}$ to this factor ($0.3\pm0.4$) resulted in a large uncertainty in the PBOA
estimated concentration.
The last three factors were related to SA, as indicated by the large contributions of secondary
species such as oxalate, $SO_4^{2-}$, MSA, and $NO_3^-$ to the factor profiles (Fig. 4). The three factors
showed different spatial and temporal contributions.
The $NO_3^-$-related SA exhibited highest contributions during winter, suggesting temperature-
driven partitioning of secondary aerosol components. Moreover the $NO_3^-$-related SA,
similarly to BB and TE, showed the highest concentrations in Vilnius, and the lowest in
Rūgšteliškis suggesting its possible relation with anthropogenic gaseous precursors (e.g.
$NO_x$).
The MSA-related SA factor manifested the highest concentrations at the marine site of Preila
during summer, and in general larger contributions during summer than winter, suggesting its
relation with marine secondary aerosol. MSA has been reported to be related to marine
secondary biogenic emissions deriving from the photo-oxidation of dimethyl sulfide (DMS)
emitted by the phytoplankton bloom occurring during the warm season (Li et al., 1993,
Crippa et al., 2013 and references therein).
The last factor ($SO_4^{2-}$-related SA) showed higher contributions during summer than in winter
without clear site-to-site variability, following the seasonal behavior of $SO_4^{2-}$ showing slightly
higher concentrations during summer than in winter, which is probably driven by the
secondary formation from gaseous photochemical reactions and aqueous phase oxidation.
This factor explained the largest part of the oxalate and $SO_4^{2-}$ variability and represented
$48\%_{avg}$ of the measured $PM_1$ by mass.
**4.4 Comparison of the source apportionment methods**
In this section we compare the offline-AMS PMF and marker-PMF results. We begin with
BBOA and TE emissions which were resolved by both approaches. The remaining OM
fraction (Other-OA) was apportioned by the offline-AMS source apportionment to B-OOA,





S-OOA and LOA (Other-OA$_{offline-AMS}$). However, the LOA contribution was statistically
significant (within 3σ) only in Vilnius during summer (Table S2), while no data were
available for these periods from the marker source apportionment. The marker source
apportionment instead attributed the Other-OA mass fraction to 4 factors (Other-OA$_{marker}$):
PBOA, as well as to $SO_4^{2-}$, $NO_3^-$, and MSA-related secondary organic aerosols (SOA, Fig.
S13). The OA concentrations of the factors retrieved from the $PM_1$ markers source
apportionment were obtained by multiplying the factor time series by the sum of the organic
markers and $OM_{res}$ contributions to the normalized factor profiles. The PM concentrations
from the marker PMF factors are displayed in Fig. 5.

### 4.4.1 Primary OA sources

Offline-AMS and marker source apportionments provided comparable BBOA estimates, with
concentrations agreeing within a 95% confidence interval (Fig. 6). Results revealed that
BBOA contributed the largest fraction to the total OM during winter in Preila and Vilnius,
while in Rūgšteliškis the largest OA source derived from B-OOA. The average winter BBOA
concentration was 1.1±0.8 µg m$^{-3}$ in Rūgšteliškis and 2±1 µg m$^{-3}$ in Vilnius (errors in this
section represent the standard deviation of the temporal variability). Overall the average
BBOA concentrations were higher at the urban background site of Vilnius and lower at the
rural terrestrial site of Rūgšteliškis. Preila showed the highes values (3±3 µg m$^{-3}$) driven by
the grass burning episode occurred at the beginning of March (Ulevicius et al., 2016).
Excluding this episode, the BBOA winter concentration was lower than in Vilnius (1.8 µg m$^{-3}$
). During winter, Preila and Vilnius showed well correlated BBOA time series ($R$=0.91).
These results highlight the important role of regional meteorological conditions on the air
quality in the south east Baltic region.
By contrast, during summer BBOA concentrations were much lower, with 40% of the points
showing statistically not significant contributions within 3σ for the offline-AMS source
apportionment and 100% for the marker source apportionment. Between late autumn and
early March the offline-AMS source apportionment revealed three simultaneous episodes
with high BBOA concentrations at the three stations, while the maker source apportionment
which is characterized by lower time resolution did not capture some of these episodes. The
first episode occurred between 19 and 25 December 2013 during a cold period with an
average daily temperature drop to -9.7 °C as measured at the Rūgšteliškis station (no



temperature data were available for the other stations). The third episode occurred between 5
and 10 March 2014 and was associated with an intense grass burning episode localized mostly
in the Kaliningrad region (Ulevicius et al., 2015, Dudoitis et al., 2015, Mordas et al., 2016).
The episode was not associated with a clear temperature drop, with the highest concentration
(14 µg m$^{-3}$) found at Preila on 10 March 2014, the closest station to the Kaliningrad region.
Similarly, at the beginning of February high BBOA concentrations were registered at the
three stations, without a clear temperature decrease. Other intense BBOA events were
detected but only on a local scale, with intensities comparable to the regional scale episodes.
Using the OM:OC$_{BBOA}$ ratio calculated from the HR water-soluble BBOA spectrum (1.88),
we estimated the BBOC$_{avg}$ concentrations during the grass burning episode (5-10 March
2014) to span between 0.8 and 7.2 µg m$^{-3}$. On a daily basis our BBOC concentrations are
consistent with the estimated ranges reported by Ulevicius et al. (2015) for non-fossil primary
organic carbon, showing also a high correlation (*R*=0.98).
TEOA estimates obtained by offline-AMS and marker-PMF agreed well with each other, with
99% of the points being not statistically different within 1σ (Fig. 6). The two approaches
confirm that TEOA is a minor source at all three stations with on average higher
concentrations in Vilnius (up to 0.8 µg m$^{-3}$), than in Preila and Rūgšteliškis (up to 0.2 µg m$^{-3}$).
Hopane concentrations were below detection limits (7 pg m$^{-3}$) for 66% of the collected
samples. TEOA, similarly to hopanes and NO$_x$, showed a clear spatial and seasonal variability
with higher concentrations in Vilnius during winter, suggesting an accumulation of traffic
emissions in a shallower boundary layer (Fig. 3b, NO$_x$ data available only for Vilnius).
During the grass burning event, we observed a peak in the total hopane concentration, and
therefore also a peak of the estimated TEOA (2.4 µg m$^{-3}$ maximum value). This relatively
high concentration is most probably not due to a local increase of TE, but rather due to a
regional transport of polluted air masses from neighboring countries (Poland and the Russian
Kaliningrad enclave). By assuming an OM:OC$_{TEOA}$ ratio of 1.2±0.1 (Aiken et al., 2008, Mohr
et al., 2008, Docherty et al., 2011, Setyan et al., 2012), we determined the corresponding
organic carbon content (TEOC). Our TEOC concentration was consistent within 1σ with the
average fossil primary OC over the whole episode estimated by Ulevicius et al. (2015),
although on a daily basis the agreement was relatively poor.
Overall, the offline-AMS source apportionment and the marker-PMF returned comparable
results for TEOA and BBOA emissions, therefore not surprisingly the two approaches yielded





OA concentrations also for the Other-OA fractions which agreed within 1σ for 90% of the
points (Figure 6). This agreement was better for Rūgšteliškis and Preila (94% and 90%,
respectively of the points not statistically different within 1σ), and worse for Vilnius (71% of
the points not statistically different within 1σ).

## 4.4.2   Other-OA sources: offline-AMS and marker-source apportionment comparison

The marker-source apportionment, in comparison to the offline-AMS source apportionment
enables resolving well-correlated sources (e.g. BBOA and $NO_3^-$-related SOA) as well as
minor sources (e.g. MSA-related SOA and PBOA) because source-specific markers were
used as model inputs. By contrast, the offline-AMS source apportionment is capable of
resolving OA sources for which no specific markers were available such as LOA, which was
separated due to the distinct spatial and temporal trends of some N-containing AMS
fragments. We first briefly summarize the Other-OA factor concentrations and their site-to-
site differences retrieved by the two techniques; subsequently we compare the two source
apportionment results.
The Other-OA$_{offline-AMS}$ factor time series are displayed in Fig. S13. The B-OOA factor
showed relatively stable concentrations throughout the year with $0.9\pm0.8_{avg}$ µg m$^{-3}$ during
summer and $1.1\pm0.9_{avg}$ µg m$^{-3}$ during winter. Although B-OOA concentrations were relatively
stable throughout the year, higher contributions were observed in Preila and Rūgšteliškis
compared to Vilnius. The extreme average seasonal concentrations were between 0.8 and 1.3
µg m$^{-3}$ at Rūgšteliškis during fall and winter, between 0.9 and 1.1 µg m$^{-3}$ at Preila during
spring and winter, and between 0.4 and 0.6 µg m$^{-3}$ in Vilnius during summer and winter.
These values do not evidence clear seasonal trends, but highlight a site-to-site variability
which will be further discussed in the following. S-OOA instead was the largest contributor to
total OM during summer with an average concentration of $1.2\pm0.8$ µg m$^{-3}$, always agreeing
between sites within a confidence interval of 95% (2 tails t-test). By contrast, during winter
the S-OOA concentration dropped to an average value of $0.3\pm0.2$ µg m$^{-3}$, with 81% of the
points not statistically different from 0 µg m$^{-3}$ within 3σ. Finally, the LOA factor showed
statistically significant contributions within 3σ only during summer and late spring in Vilnius.
Despite its considerable day-to-day variability this fraction contributed $1.0\pm0.8$ µg m$^{-3}_{avg}$ in
Vilnius during summer.




The markers source apportionment instead attributed 85%$_{avg}$ of the Other-OA$_{marker}$ mass to the
SO$_4^{2-}$-related SOA, while NO$_3^-$-related SOA, MSA-related SOA, and PBOA explained
respectively 9%$_{avg}$, 5%$_{avg}$ and 1%$_{avg}$ of the Other-OA$_{marker}$ mass (Fig. S13). The SO$_4^{2-}$-related
SOA average concentration was 2.4 µg m$^{-3}$ during summer and 1.7 µg m$^{-3}$ during winter with
no significant differences from station to station, suggesting a regional origin of the factor.
The NO$_3^-$-related SOA concentration was 0.4 µg m$^{-3}$$_{avg}$ during winter, only 0.03$_{avg}$ µg m$^{-3}$,
during summer, corresponding to 10%$_{avg}$ and 1% of the OA, respectively. Moreover, the NO$_3^-$
-related SOA during winter showed the highest average concentrations in Vilnius with 0.5 µg
m$^{-3}$ and the lowest in Rūgšteliškis with 0.3 µg m$^{-3}$$_{avg}$. The MSA-related SOA instead
manifested the highest concentrations during summer with an average of 0.12 µg m$^{-3}$$_{avg}$.
Higher values were observed during summer at the rural coastal site of Preila where the
average concentration was 0.28 µg m$^{-3}$$_{avg}$ corresponding to 10%$_{avg}$ of the OM. Finally, the
PBOA factor exhibited the largest seasonal concentrations during spring at the rural terrestrial
site of Rūgšteliškis with an average of 0.05 µg m$^{-3}$$_{avg}$, while the summer average
concentration was 0.02 µg m$^{-3}$ consistent with the low PBOA estimates reported in Bozzetti et
al. (2016) for the submicron fraction during summer.
Many previous studies reported a source apportionment of organic and inorganic markers
concentrations (Viana et al., 2008 and references therein). In these studies SO$_4^{2-}$, NO$_3^-$, and
NH$_4^+$ were typically used as tracers for secondary aerosol factors commonly associated with
regional background and long-range transport; here we compare the apportionment of the
SOA factors obtained from the marker source apportionment and the OOA factors separated
by the offline-AMS source apportionment. Moreover, contrasting the two source
apportionments may provide insight into the origin of the OOA factors retrieved from the
offline-AMS source apportionment, and into the origin of the SOA factors resolved by the
offline-AMS source apportionment. To our knowledge an explicit comparison has not yet
been reported in the literature.
Table 2: Pearson correlation coefficients between Other-OA components from offline-AMS
and marker-source apportionment.

| | | Other-OA$_{marker}$ | | | |
|---|---|---|---|---|---|
| | | SO$_4^{2-}$-related SOA | MSA-related SOA | NO$_3^-$-related SOA | PBOA |
| Other-OA$_{offline-}$ | LOA | 0.33 | 0.16 | -0.08 | 0.10 |
| | B-OOA | 0.70 | 0.22 | 0.21 | 0.47 |





| | | | | | |
|---|---|---|---|---|---|
| AMS | S-OOA | 0.60 | 0.45 | -0.47 | 0.05 |

Table 2 reports the correlations between the time series of the Other-OA$_{marker}$ factors and the
Other-OA$_{offline-AMS}$ factors (Figs. 6 and S13). These correlations are mostly driven by seasonal
trends as none of these sources shows clear spikes except for LOA during summer in Vilnius.
Using the correlations coefficients we can identify the mostly related factors from the two
source apportionments.
The $SO_4^{2-}$-related SOA explained the largest fraction of the Other-OA$_{marker}$ mass (85%$_{avg}$),
and it was the only Other-OA$_{marker}$ factor always exceeding the individual concentrations of
B-OOA and S-OOA, indicating that the variability explained by the $SO_4^{2-}$-related SOA in the
marker-source apportionment is explained by both OOA factors in the offline-AMS source
apportionment. Moreover, the $SO_4^{2-}$-related SOA seasonality seems consistent with the sum
of S-OOA and B-OOA with higher concentrations in summer than in winter. This observation
suggests that the OOA factors resolved by offline-AMS are mostly of secondary origin and
the $SO_4^{2-}$-related SOA, typically resolved by the markersource apportionment, explains the
largest fraction of the OOA factors apportioned by offline-AMS which includes both biogenic
SOA and aged background OA.
The $NO_3^-$-related SOA and the PBOA were mostly related to the B-OOA factor as they
showed higher correlations with B-OOA than with S-OOA. The B-OOA factor therefore may
explain a small fraction of primary sources (PBOA), which however represents only 0.6%$_{avg}$
of the total OA.
The MSA-related SOA showed the highest correlation with the S-OOA factor, as the two
sources exhibited the highest concentrations during summer, although the MSA-related SOA
preferentially contributed at the rural coastal site of Preila. While we already discussed the
probable secondary biogenic origin of S-OOA, the correlation with the MSA-related SOA
suggests that the S-OOA factor, especially at the rural coastal site of Preila, explains also a
large fraction of the marine biogenic SOA. Assuming all the MSA-related SOA to be
explained by the S-OOA factor, we estimate a marine biogenic SOA contribution to S-OOA
of 27%$_{avg}$ during summer at Preila, while this contribution is lower at the other stations
(12%$_{avg}$ in Rūgšteliškis during summer, 7% in Vilnius during spring, no summer data for
Vilnius Fig. S13). As already mentioned, here we assume all the MSA-related SOA to be
related to marine secondary biogenic emissions, however other studies also report MSA from



terrestrial biogenic emissions (Jardine et al., 2015), moreover a certain fraction of the MSA-
related SOA can also be explained by the B-OOA factor. Overall these findings indicate that
the terrestrial sources dominate the S-OOA composition, nevertheless the marine SOA
sources may represent a non-negligible fraction, especially at the marine site.
Another advantage obtained in coupling the two source apportionment results is the
possibility to study the robustness of the factor analyses by evaluating the consistency of the
two approaches as we already discussed for the primary OA and Other-OA fractions. By
subtracting LOA and S-OOA from Other-OA$_{marker}$ we can estimate the equivalent B-OOA
concentration from the marker source apportionment (B-OOA$_{marker}$). Unlike the B-OOA
factor from offline-AMS, whose contribution is lower at Vilnius, B-OOA$_{marker}$ did not show
statistically different concentrations at all stations within a confidence interval of 95%. This
discrepancy could indicate some PMF residual uncertainties or biases not considered in our
error estimate for offline-AMS and/or markers source apportionments for Vilnius, which
could not be detected without coupling the 2 source apportionment approaches.
**4.5 $f$CO$^+$ vs. $f$CO$_2^+$**
Figure 7 displays the water-soluble $f$CO$^+$ vs. $f$CO$_2^+$ scatter plot. A certain correlation ($R$=0.63)
is seen, with $f$CO$^+$ values being systematically lower than $f$CO$_2^+$ (CO$_2^+$:CO$^+$: 1$^{st}$ quartile 1.50,
median 1.75, 3$^{rd}$ quartile 2.01), whereas a 1:1 CO$_2^+$:CO$^+$ ratio is assumed in standard
AMS/ACSM analyses (Aiken et al., 2008; Canagaratna et al., 2007). Comparing the measured
CO$_2^+$:CO$^+$ values for the bulk WSOM and for pure gaseous CO$_2$ might provide insight into
the origin of the CO$^+$ fragment in the AMS. The fragmentation of pure gaseous CO$_2$ returned
a CO$_2^+$:CO$^+$ ratio of 8.21$_{avg}$ which is significantly higher than our findings for the water-
soluble bulk OA (1.75$_{med}$) suggesting that the WSOM decarboxylation on the AMS vaporizer
represents only a minor source of CO$^+$.
Figure 7b and Fig. 8 show that not only does the water-soluble (WS) CO$_2^+$:CO$^+$ ratio
systematically differ from 1, but it also varies throughout the year with higher CO$_2^+$:CO$^+$
values associated with warmer temperatures (Fig. 7b). The lower CO$_2^+$:CO$^+$ ratios in winter
are primarily due to BB, as the WSBBOA factor profile showed the lowest CO$_2^+$:CO$^+$ ratio
(1.20$_{avg}$) among all the apportioned WS factors (2.00$_{avg}$ for B-OOA, 2.70$_{avg}$ for S-OOA, and
2.70$_{avg}$ for LOA). We observed a seasonal variation of the CO$_2^+$:CO$^+$ ratio also for the water-





soluble OOA (S-OOA + B-OOA) mass spectral fingerprint. The $CO_2^+$:$CO^+$ ratio was slightly
lower for B-OOA than for S-OOA ($2.00_{avg}$ for B-OOA, 2.70 for S-OOA). Nevertheless, given
the low S-OOA relative contribution during winter (Fig. 3), we note that the total OOA
showed a slightly lower $CO_2^+$:$CO^+$ ratio during winter than in summer (Fig. S14), indicating
that the OOA mass spectral fingerprint evolves over the year, possibly because of different
precursor concentrations, and different photochemical activity.
Fig. 7a shows that most of the measured {$fCO^+$;$fCO_2^+$} combinations lies within the triangle
defined by the BBOA, S-OOA and B-OOA {$fCO^+$;$fCO_2^+$} combinations. The LOA factor
{$fCO^+$;$fCO_2^+$} combination lies within the triangle as well, but is anyways a minor source and
thus unlikely to contribute to the $CO_2^+$/$CO^+$ variability. We parameterized the $CO^+$ variability
as a function of the $CO_2^+$, and $C_2H_4O_2^+$ fragment variabilities using a multi-parameter fit
according to Eq. (8). $CO_2^+$ and $C_2H_4O_2^+$ were chosen as B-OOA and BBOA tracers,
respectively, with B-OOA and BBOA being the factors that explained the largest fraction of
the $fCO^+$ variability (85% together).

$$CO^+_i = a \cdot CO_2^+{}_i + b \cdot C_2H_4O_2^+{}_i \qquad (8)$$

Although this parameterization is derived from the WSOM fraction $CO_2^+$, $C_2H_4O_2^+$, and $CO^+$
originate from the fragmentation of oxygenated, i.e. mostly water-soluble compounds.
Accordingly, this parameterization might also well represent the total bulk OA (as the offline-
AMS recoveries of these oxygenated fragments are relatively similar: $R_{CO_2^+}$=0.74,
$R_{C_2H_4O_2^+}$=0.61, Daellenbach et al., 2016). Note that this parameterization may represent very
well the variation of $CO^+$ in an environment impacted by BBOA and OOA, but should be
used with caution when other sources (such as COA) may contribute to $CO^+$, $CO_2^+$ and
$C_2H_4O_2^+$. In order to check the applicability of this parameterization to a PMF output, we
recommend monitoring the $CO_2^+$ and $C_2H_4O_2^+$ variability explained by the OOA and BBOA
factors. In case a large part of the $CO_2^+$ and $C_2H_4O_2^+$ variability is explained by OOA and
BBOA, the parameterization should unlikely return uncertain $CO^+$ values. The coefficients $a$
and $b$ of Eq. (8) were determined as 0.52 and 1.39 respectively, while the average fit residuals
were estimated to be equal to 10% (Fig. S15). In contrast, parameterizing $CO^+$ as proportional
to $CO_2^+$ only (as done in the standard AMS analysis scheme with coefficients updated to the
linear fit between $CO^+$ and $CO_2^+$ (1.75)) yielded $20\%_{avg}$ residuals, indicating that such a
univariate function describes the $CO^+$ variation less precisely.





An alternative parameterization is presented in the SI in which the contribution of moderately
oxygenated species (such as S-OOA) to $CO^+$ was also considered by using $C_2H_3O^+$ as an
independent variable. We show that the dependence of $CO^+$ on $C_2H_3O^+$ is statistically
significant (Fig. 7b) as also suggested by the PMF results (S-OOA contributes 12% to the
$CO^+$ variability). However, the parameter relating $CO^+$ to $C_2H_3O^+$ is negative, because the
$CO^+:CO_2^+$ and $CO^+:C_2H_4O_2^+$ ratios are lower in moderately oxygenated species compared to
species present in BBOA and B-OOA. While this parameterization captures the variability of
$CO^+$ across the seasons better compared to a 2-parameter fit for the present dataset, it may be
more prone to biases in other environments due to the known contributions of other factors to
$C_2H_3O^+$. For example, cooking-influenced organic aerosol (COA) often accounts for a
significant fraction of $C_2H_3O^+$. For ambient datasets we propose the use of $CO_2^+$ and $C_2H_4O_2^+$
only, which may capture less variation but is also less prone to biases. Although our results
suggest that the available $CO^+$ and O:C estimates (Aiken et al., 2008; Canagaratna et al.,
2015) may not well capture the $CO^+$ variability, our $CO^+$ parameterization should not be
applied to calculate the O:C ratios or recalculate the OA mass from AMS datasets, as those
are calibrated assuming a standard fragmentation table (i.e. $CO_2^+ = CO^+$).
In a recent work, Canagaratna et al. (2015) reported the Ar nebulization of water soluble
single compounds to study the HR-AMS mass spectral fingerprints in order to improve the
calculation of O:C and OM:OC ratios. Following the same procedure, we nebulized a subset
of the same standard compounds including malic acid, azalaic acid, citric acid, tartaric acid,
cis-pinonic acid, and D(+)-mannose. We obtained comparable $CO_2^+:CO^+$ ratios (within 10%)
to those of Canagaratna et al. (2015) for all the analyzed compounds, highlighting the
comparability of results across different instruments. With the exception of some
multifunctional compounds, the water-soluble single compounds analyzed by Canagaratna et
al. (2015) mostly showed $CO_2^+:CO^+$ ratios <1, systematically lower than the $CO_2^+:CO^+$ ratios
measured for the bulk WSOM in Lithuania (1st quartile 1.50, median 1.75, 3rd quartile 2.01),
which represents a large fraction of the total OM (bulk EE: median = 0.59, 1st quartile = 0.51,
3rd quartile = 0.72). This indicates that the selection of appropriate reference compounds for
ambient OA is non-trivial, and the investigation of multifunctional compounds is of high
importance.


## 5 Conclusions

PM$_1$ filter samples were collected over an entire year (November 2013 to October 2014) at three different stations in Lithuania. Filters were analyzed by water extraction followed by nebulization of the liquid extracts and subsequent measurement of the generated aerosol with an HR-ToF-AMS (Daellenbach et al., 2016). For the first time, the nebulization step was conducted in Ar, enabling direct measurement of the CO$^+$ ion, which is typically masked by N$_2^+$ in ambient air and assumed to be equal to CO$_2^+$ (Aiken et al., 2008). CO$_2^+$:CO$^+$ values >1 were systematically observed, with a mean ratio of 1.7±0.3. This is likely an upper limit for ambient aerosol, as only the water-soluble OM fraction is measured by the offline-AMS technique. CO$^+$ concentrations were parameterized as a function of CO$_2^+$, and C$_2$H$_4$O$_2^+$, and this two-variable parameterization showed a superior performance to a parameterization based on CO$_2^+$ alone, because CO$^+$ and CO$_2^+$ show different seasonal trends.

PMF analysis was conducted on both the offline-AMS data described above and a set of molecular markers together with total OM. Biomass burning was found to be the largest OM source in winter, while secondary OA was largest in summer. However, higher concentrations of primary anthropogenic sources (traffic and biomass burning) were found at the urban background station of Vilnius. The offline-AMS and marker-based analyses also identified local emissions and primary biological particles, respectively, as factors with low overall but episodically important contributions to PM. Both methods showed traffic exhaust emissions to be only minor contributors to the total OM; which is not surprising given the distance of the three sampling stations from busy roads.

The two PMF analyses apportioned SOA to sources in different ways. The offline-AMS data yielded factors related to regional background (B-OOA) and temperature-driven (likely biogenic-influenced) emissions (S-OOA), while the marker-PMF yielded factors related to nitrate, sulfate, and MSA. For the offline-AMS PMF, S-OOA was the dominant factor in summer and showed a positive exponential correlation with the average daily temperature, similar to the behavior observed by Leaitch et al. (2011) in a Canadian boreal forest. Combining the two source apportionment techniques suggests that the S-OOA factor includes contributions from both terrestrial and marine secondary biogenic sources, while only small PBOA contributions to submicron OOA factors are possible. The analysis highlights the importance of regional meteorological conditions on air pollution in the southeastern Baltic region, as evidenced by simultaneous high BBOA levels at the three stations during three





different episodes in winter and by statistically similar S-OOA concentrations across the three
stations during summer.

## Acknowledgements

The research leading to these results received funding from the Lithuanian–Swiss
Cooperation Programme "Research and Development" project AEROLIT (Nr. CH-3-ŠMM-
01/08). JGS acknowledges the support of the Swiss National Science Foundation (Starting
Grant No. BSSGI0 155846). IE-H acknowledges the support of the Swiss National Science
Foundation (IZERZ0 142146).





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

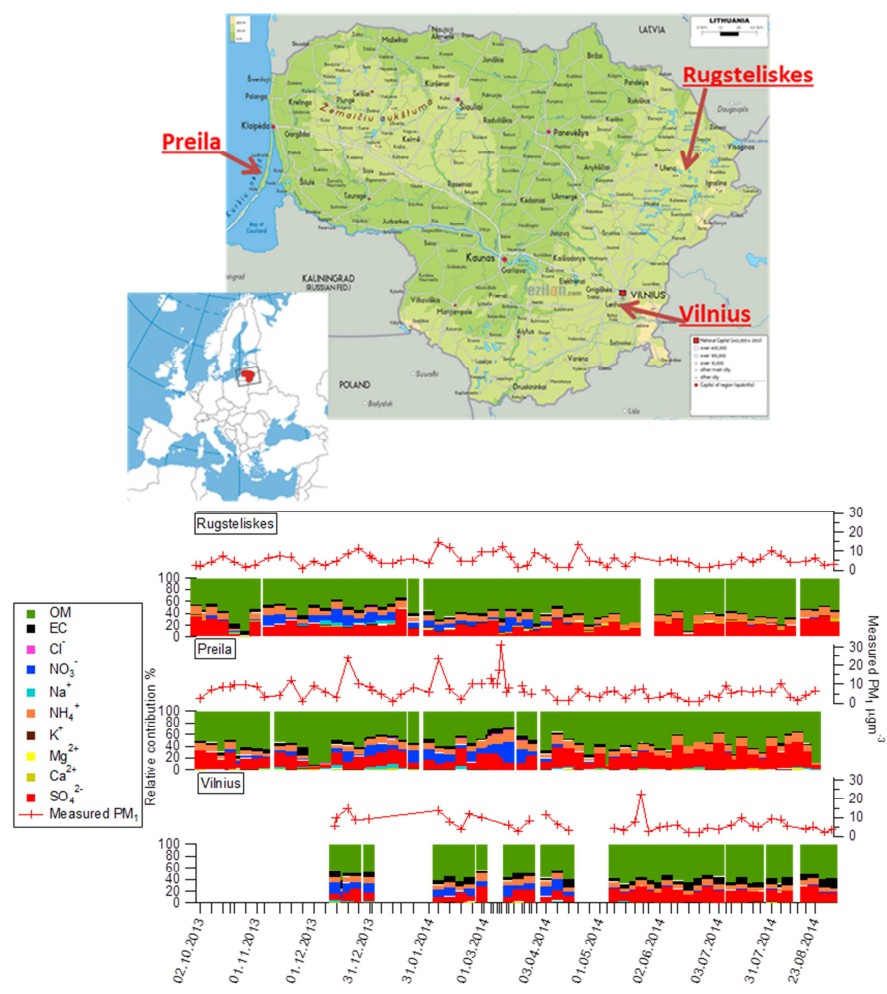

3    Figure 1. Sampling locations, and measured PM$_1$ composition.





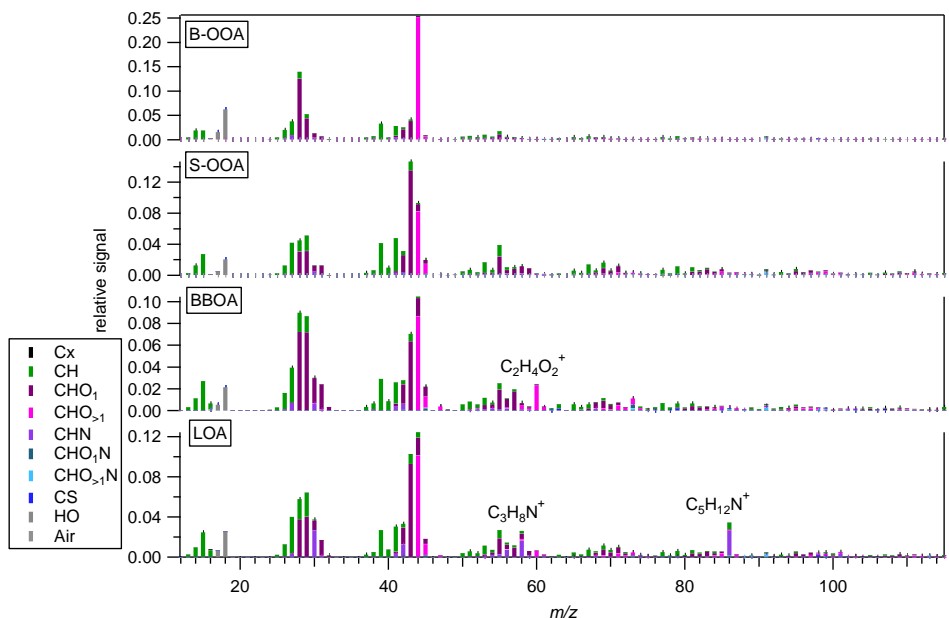

2    Figure 2. Offline-AMS PMF factor profiles.

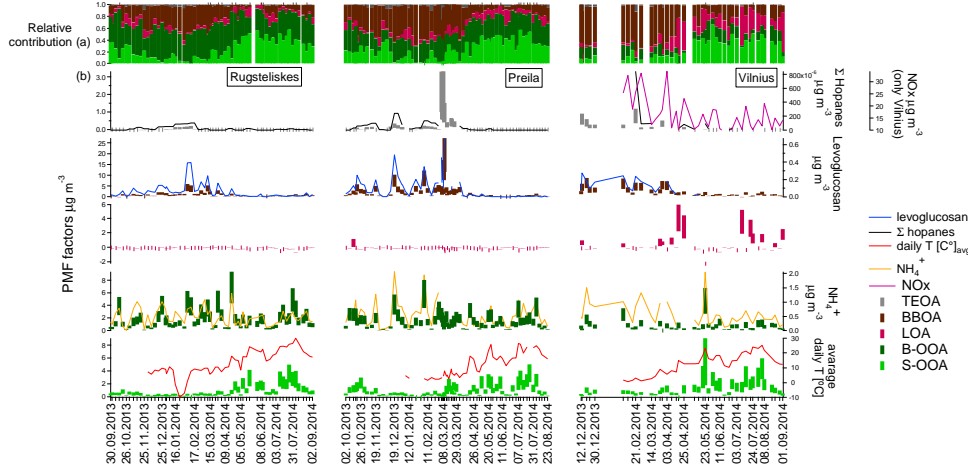

5    Figure 3. a) Temporal evolutions of relative contributions to the OA factors; b) OA sources

6    and corresponding tracers: concentrations and uncertainties (shaded areas).



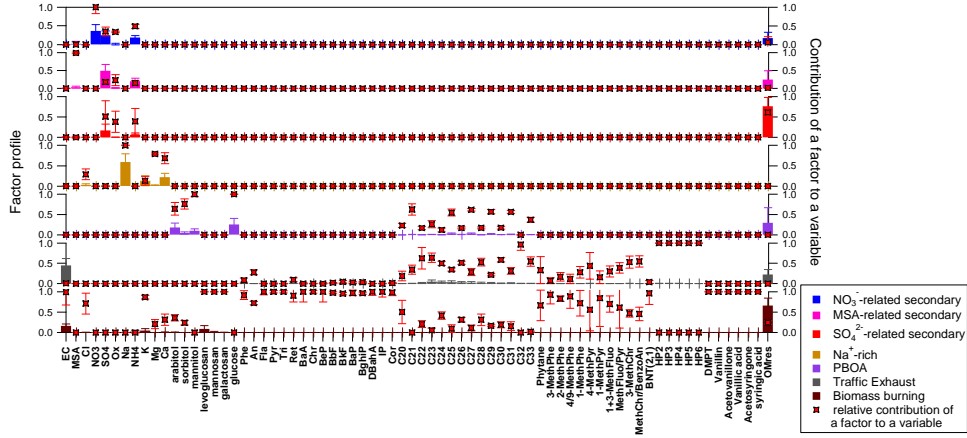

Figure 4. Marker-PMF factor profiles (bars) and relative contributions of the factors to the
measured variables (symbols).

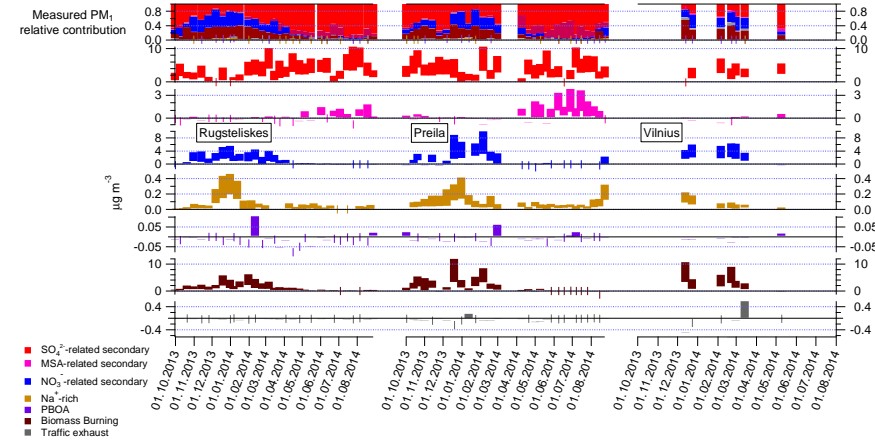

Figure 5. PM$_1$ marker source apportionment: factor time series and relative contributions.
Shaded areas indicate uncertainties (standard deviation) of 20 bootstrap runs.



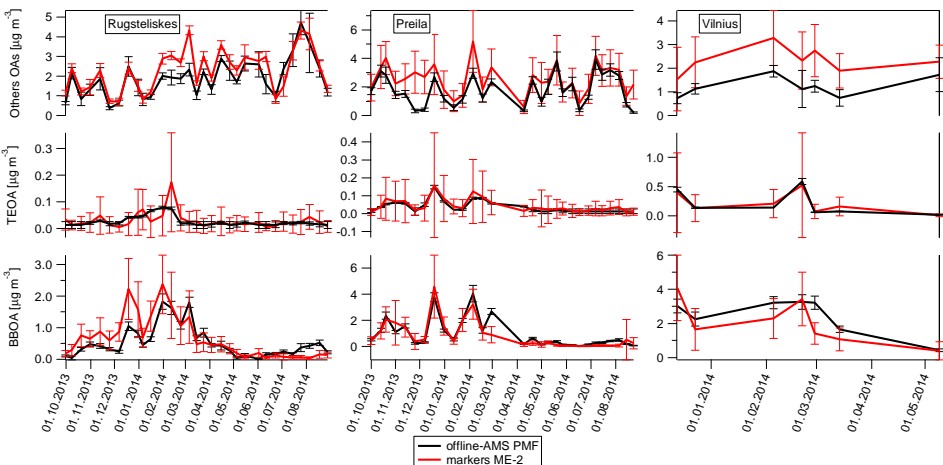

Figure 6. Marker-PMF and offline-AMS OM source apportionment comparison.

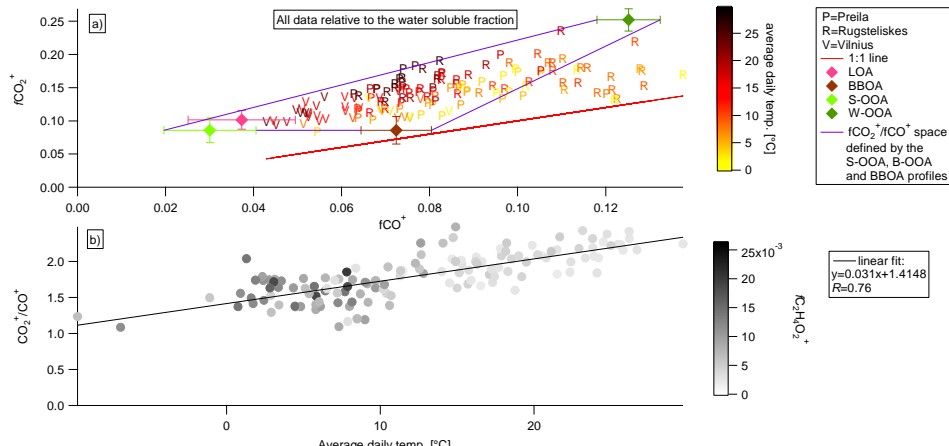

Figure 7. a) water-soluble $f$CO$_2^+$ vs $f$CO$^+$ scatter plot. Color code denotes the average daily
temperature [°C], diamonds indicate the $f$CO$_2^+$/$f$CO$^+$ ratio for different PMF factor profiles.
The 1:1 line is displayed in red. Few points from Rūgšteliškis lie outside the triangle,
suggesting they are not well explained by our PMF model. However, Fig. S5 displays flat
residuals for Rūgšteliškis, indicating an overall good WSOM explained variability by the
model. b) Scatter plot of the water-soluble CO$^{2+}$ to CO$^+$ ratio *vs.* average daily temperature.
Grey code denotes $f$C$_2$H$_4$O$_2^+$.





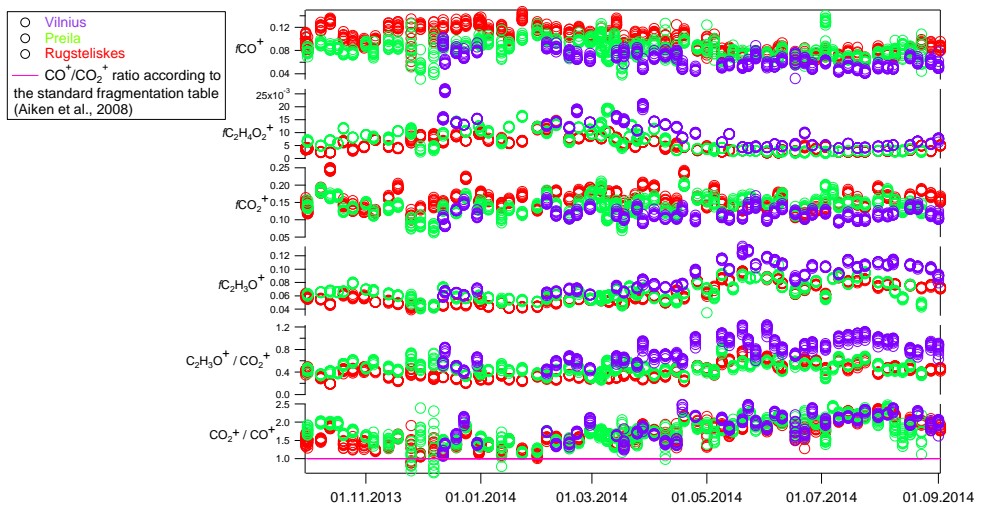

2    Figure 8. Time-dependent fractional contributions (*f*) of typical AMS tracers.

