# Peer review of "Argon offline-AMS source apportionment of organic"

_Atmospheric Chemistry and Physics, 2016_

## Referee Comment (RC1) · Anonymous Referee #1 · 20 Jun 2016

General Comments:

This manuscript presents an analysis of the composition and source apportionment of PM1 filters collected at three sites in Lithuania. For this offline technique, the aqueous extracts from filters were nebulized with Ar for introduction into the HR-ToF-AMS. The use of Ar as the nebulization gas enabled an analysis of the $CO+/CO2+$ fragment ratio and trends in that ratio with season. Positive matrix factorization was also applied on both the offline AMS data set as well as an offline marker data set collected using the same filters. This manuscript provides a good demonstration of the type of data sets that can be generated via this offline AMS technique and the $CO+/CO2+$ analysis provides new insights into the interpretation of AMS data from ambient samples. Thus,

[Figure]

I see this paper as appropriate for publication in ACP. However, I have a few concerns, mostly related to sampling artifacts that need to be addressed prior to publication.

Specific comments:

P2 L9: Traffic exhaust OA is listed as a PMF factor from AMS spectra, yet in the experimental it is noted that the contribution is too low to be resolved with PMF and is instead estimated using a CMB approach. I suggest rewording the abstract to clarify this.

P5 L24: The nebulizer used was operated at 60 °C, how long are the aerosols in this heated region? Was this temperature in the nebulizer also used in the Daellenbach et al. analysis? What effect might this high temperature have on the composition of the organics measured with the AMS compared to online analysis? If this temperature was not used for the Daellenbach analysis, what effect might this have on the factor specific recoveries of this work compared to the results from that previous analysis?

P18 L25: PM1 composition discussed here and shown in Figure 1 shows ions that can be measured with both the AMS and IC (e.g. SO4, NO3, etc.). Do the contributions shown in Figure 1 correspond to the IC measurements or AMS? For ions that can be quantified with both techniques, how do the values compare between the AMS and IC?

P19 L14-20: The nitrate concentration shows clear seasonality with larger contributions in the winter and the sulfate concentration looks relatively constant throughout the year. However, in Figure 1, the ammonium concentration appears to also be relatively constant throughout the year. Is this correct? If so, can the authors comment on potential counter ions for NO3 ?

P20 L 28-31: The background-OOA factor appears to correlate with NH4+ much better at Preila and Vilnius than Rugsteliskes (Figure S11). Are there any potential reasons for the lower apparent correlation at Rugsteliskes? How much uncertainty is there in the NH4+ measurement? What is the significance of a correlation of B-OOA with

NH4+?

Section 2.1 and P21 L1-17: Were the High-Volume samplers located in temperature controlled rooms? If not, what effect could higher summer temperatures have on the composition of the organic compared to the winter samples? Could the S-OOA factor be complicated by collection differences caused by the loss (on the filter) of more volatile organic molecules during summer months?

Technical corrections:

P2 L6: the CO2+:CO+ ratios reported in section 4.5 are greater than 1. The less than sign should be switched.

P10 L22-23: a verb such as "was used" is missing.

P22 L3: I suggest some mention directing the reader to Figure 5 be made in the text as the time series for the factors are discussed in this section but no mention of Figure 5 is made.

P25 L13: "Using the ratio (1.88) calculated from offline-AMS". Suggest adding OM/OCBBOA ratio to communicate what ratio is being used in the calculation here.

P30 L 25-26: suggest rephrasing, the double negative "unlikely return uncertain CO+ values" is confusing.

P45 Figure 2 and P46 Figure 4: Suggest either writing out the factor names in the labels (background-OOA instead of B-OOA etc.) or giving the names and labels in the caption.

---

## Referee Comment (RC2) · Anonymous Referee #2 · 30 Jun 2016

Review of "Argon offline-AMS source apportionment of organi aerosol over yearly cycles for an urban, rural and marine site in Northern Europe" by Bozzetti et al.

This manuscript reported an analysis of $PM_1$ compositions and sources at three different sites in Lithuania based on filter samples. The authors applied AMS and other instruments to analyze the filter samples, and then performed PMF analysis to study the sources of OA and $PM_1$. This study presented a method/case to study the sources of total ambient OA based on the measurements of water soluble OA only. That is, apply PMF analysis on the water soluble organic mass spectra, identify multiple factors, and rescale the water soluble concentration to total concentration by applying recovery ratios. This is an interesting method but has large uncertainties, which arise from the recovery ratio. I think this manuscript is suitable for publication in ACP once the following comments have been addressed.

**Major comments**

1.      Ambient total OA source apportionment based on the measurement of water soluble OA.

        The major uncertainty of this method arises from the recovery ratio (Rz), which is a reflection of the bulk extraction efficiency and water solubility of OA factors. It is not clear how the Rz values are obtained in this study. As I understand, the authors randomly selected Rz from Daellenbach et al. (2016) as initial conditions and fit Eq. (6) to get $R_{LOA}$. If so, how are $R_{BBOA}$ and $R_{OOA}$ obtained? Why are they different from the values in Daellenbach et al. (2016). Also, it is not clear which Rz values are eventually applied, from Daellenbach et al. (2016) or the values calculated in this study?

        The authors mentioned that the bulk extraction efficiency in this study is lower than that in Daellenbach et al. (2016). This result is not surprising since one OA factor likely has contribution from multiple sources and the water solubility of OA factors may vary with site and season. For example, the water solubility of BBOA ranges from 64% to 80% (Sciare et al., 2011; Timonen et al., 2008). In addition, this method is not sensitive to primary OA factors (e.g., HOA and Cooking OA), which is largely water insoluble. This is another reason why HOA cannot be resolved from the PMF analysis. The limitations should be better discussed in the manuscript.

What suggestions do the authors have for researchers who want to use the method as proposed in this manuscript? For example, should they follow the same filter extraction procedures as in this study? How to calculate the Rz?

2.      Discussions on instruments comparison are required.

(1) Inorganic ions such as $NH_4^+$, $NO_3^-$, and $SO_4^{2-}$ are measured by both AMS and IC. The authors should present the instruments comparison.

(2) Page 9 Line 29-30. The AMS measured concentration is scaled to match the WSOC measurement. What's the scale ratio? Is the scale ratio the same for all filter samples?

3.      The difference in separation and classification of OA factors between online and offline-AMS (Page 20 Line 14-27).

I disagree with the statement that "online-AMS OOA factors are commonly classified based on their volatility", because chemistry and sources also affect the factor separation. For example, the separation of IEPOX-OA factor (Budisulistiorini et al., 2013; Hu et al., 2015) or called isoprene-OA factor (Xu et al., 2015) is driven by IEPOX chemistry, but not volatility. Also, Xu et al. (2015) showed that nighttime monoterpene oxidation by nitrate radical contributes to less-oxidized OOA (as termed SV-OOA in this study).

The authors stated that "the offline-AMS sources apportionment separates factors by seasonal trends rather than volatility". However, sometimes, seasonal trend affects the source apportionment through volatility. For example, Page 23 Line 26-27 discussed that higher $NO_3^-$-related SA exhibits higher concentration in winter than summer, which is due to the semi-volatile nature of $NO_3^-$ (Page 19 Line 20).

4.      OM/OC ratio.

In this study, the OM/OC is calculated by Aiken method (Page 12 Line 20). However, a recent study by Canagaratna et al. (2015) improved the estimation from Aiken method by including composition-dependent correction factors. The Canagaratna method is recommended to use. Since many calculations in this study depend on the OM/OC ratio, how would it affect the results/conclusions if the authors use Canagaratna method to calculate the OM/OC ratio?

5.      Background-OOA (B-OOA) factor.

When the authors selected solutions, one criterion is the correlation between B-OOA and $NH_4^+$ (Page 12 Line 8). The authors should explain the use of $NH_4^+$. $SO_4^{2-}$ is regional and usually used as background OA. What's the correlation between B-OOA and $SO_4^{2-}$? In Page 20 Line 30,

it is stated that B-OOA correlates well with $NH_4^+$. However, the correlation between B-OOA and $NH_4^+$ varies with site as shown in Fig. S11. For example, the correlation is really weak for the Rugsteliskis site.

If B-OOA represents background OA, why is B-OOA lower in urban site than the other sites? I disagree with the authors' argument that this difference is caused by PMF residual uncertainties or biases (Page 29 Line 10). The authors' argument is flawed because it is based on circular assumptions. When the authors calculate B-OOA$_{marker}$, the LOA and S-OOA are based on PMF analysis **without** considering "some residual uncertainties or biases". If the authors considered "some residual uncertainties or biases" and re-performed PMF analysis, the concentrations of LOA and S-OOA would change, which would influence and concentration of B-OOA$_{marker}$. In that circumstance, B-OOA$_{offline-AMS}$ may agree among all three sites, but B-OOA$_{marker}$ may be different among all three sites.

**Minor comments**

1. TEOA is resolved from CMB, not PMF. This needs to be clarified in multiple places in the manuscript, such as Page 2 Line 9 and Page 23 Line 30. Considering that the TEOA concentration is small and only one filter has statistical significant TEOA concentration (Page 22 Line 27), I suggest the authors to remove the comparison about TEOA concentration between sites (for example, Page 32 Line 15-17).

2. Page 2 Line 10. Please rephrase to "two oxygenated OA factors, summer OOA (S-OOA) and background OOA (B-OOA)".

3. Page 2 Line 16 vs. Line 18. Use OA or OM. Be consistent.

4. Page 4 Line 3. Please rephrase to "source apportionment on the submicron water soluble OA" in order to be precise about the method.

5. Page 5 Line 24. The nebulizer temperature is 60°C, which is different from Daellenbach et al. (2016). Also, the nebulizer system in this study is different from that in Daellenbach et al. (2016). Would these differences cause the difference in Rz between studies?

6. Page 5 Line 27-28. The correction of blank is not appropriate. This is because the particles generated from nebulizing DI water only are too small to be detected by AMS. However, the organics associated with DI water will be detected by AMS when nebulizing real filter extracts because the particles are big. I suggest the authors to nebulize ammonium sulfate solution (i.e.,

dissolve ammonium sulfate in DI water with similar concentration as ambient filters) and use the detected organic concentration as blank.

7.      Page 9 Line 7-9. Although the detailed procedures have been discussed in Daellenbach et al. (2016), it is still helpful to briefly discuss the method in the manuscript, especially how the recovery ratios are calculated.

8.      Page 10 Line 28. Please rephrase to "this factor has too small contribution in the water extracts to be resolved".

9.      Page 12 Line 6. This sentence has been repeated twice. Delete.

10.     Page 12 Line 13-16. AMS measures OM, instead OC. Please be clear that the conversion from OM to OC is for the carbon mass closure in Eq. (6).

11.     Page 12 Eq. (6). WSW-OOA should be WSB-OOA. Is Rz the same for S-OOA and B-OOA since the same $R_{OOA}$ is applied for both factors?

12.     Page 14 Line 20. What's the $OM_{res}/OM$ ratio?

13.     Page 15 Line 21. List the non-source specific variables.

14.     What's the $Hopanes_{sum}/OC$ ratio in the traffic exhaust factor? Is it consistent with the CMB method (i.e., 0.0012 in Page 11 Line 15)?

15.     Page 16 Line 25. Should be "$EC/OM_{res}$" ratio.

16.     Page 17 Line 10-16. The discussion is not clear. Suggest re-wording.

17.     Page 20 Line 1-3. List the levoglucosan/BBOC range in the literature. Similar suggestions for other places. For example, list the non-fossil primary organic carbon in Page 25 Line 13 and average fossil primary OC in Page 25 Line 29.

18.     Page 21 Line 2. I disagree with that S-OOA increases exponentially with average daily temperature from the data points in this study (Fig. S12). For example, many data points with T > 25°C do not have high S-OOA concentration and do not follow the exponential fit.

19.     Page 22 Line 13-15. This has been mentioned previously in Page 20 Line 1-3. It is not proper to discuss BBOC here because this section focuses on the marker-PMF, instead of offline AMS. Similar problem for Page 22 Line 23-24.

20.     Page 23 Line 14-15. The observation that nitrate concentration is higher in urban site than rural site has been shown in many previous studies (Xu et al., 2016; McMeeking et al., 2012), which should be cited here.

21.     Page 23 Line 30-31. This sentence is confusing. The remaining OM fraction is termed as $OM_{res}$ in Page 10 Line 20, but termed as Other-OA here. It should be clearly stated that Other-OA refers to OA after excluding BB and TE.

22.     Page 24 Line 18. Should be "higher"

23.     Page 24 Line 21-23. (1) Which method did the authors use to get the BBOA concentration and correlation in this sentence? (2) It would be helpful to include a scatter plot between Preila and Vilnius. (3) I disagree with "the importance of regional meteorological conditions" as stated in this sentence and Page 32 Line 31-32. Firstly, the BBOA concentrations are different between two sites. Secondly, the BBOA in the Rugsteliskis site does not correlate with the other two sites.

24.     Page 24 line 29. Both methods have the same time resolution (one filter per day).

25.     Page 25 line 15. In the statistical significance test, why is sometimes $1\sigma$ is used but sometimes $3\sigma$ is used (for example, Page 26 Line 28).

26.     Page 26 Line 30. Should be "factor" instead of "fraction".

27.     Table 2. The correlation coefficient R between $NO_3$-related SOA and B-OOA is only 0.21. Thus, it is not meaningful to discuss the relationship between $NO_3$-related SA and B-OOA (Page 28 Line 17). Similar problem for the relationship between MSA-related SOA and S-OOA (Page 28 Line 21).

28.     Page 29 Line 18. Please rephrase to "$f_{CO2}$ value is higher than $f_{CO}$".

29.     Page 29 Line 24-25. The logic is not clear. Why does higher $CO_2^+/CO^+$ ratio of gas CO2 suggest a minor contribution from WSOM decarboxylation to $CO^+$.

30.     Page 30 Line 7. Many data points from the Rugsteliskis site are outside the triangle range in Fig. 7a.

31.     Page 31 Line 4. The correlation between $CO^+$ and $C_2H_3O^+$ is not shown in Fig. 7b. It would be helpful to show a scatter plot.

32.     Page 31 Line 16. Canagaratna et al. (2015) carefully discussed the $CO_2^+/CO^+$ ratio of a number of standards, which should be discussed and mentioned more in the manuscript.

33.     Figure 5. The grey caps of traffic exhaust are not clear in this figure.

Reference

Budisulistiorini, S. H., Canagaratna, M. R., Croteau, P. L., Marth, W. J., Baumann, K., Edgerton, E. S., Shaw, S. L., Knipping, E. M., Worsnop, D. R., Jayne, J. T., Gold, A., and Surratt, J. D.: Real-Time Continuous Characterization of Secondary Organic Aerosol Derived from Isoprene Epoxydiols in Downtown Atlanta, Georgia, Using the Aerodyne Aerosol Chemical Speciation Monitor, Environ Sci Technol, 47, 5686-5694, Doi 10.1021/Es400023n, 2013.

Canagaratna, M. R., Jimenez, J. L., Kroll, J. H., Chen, Q., Kessler, S. H., Massoli, P., Hildebrandt Ruiz, L., Fortner, E., Williams, L. R., Wilson, K. R., Surratt, J. D., Donahue, N. M., Jayne, J. T., and Worsnop, D. R.: Elemental ratio measurements of organic compounds using aerosol mass spectrometry: characterization, improved calibration, and implications, Atmos. Chem. Phys., 15, 253-272, 10.5194/acp-15-253-2015, 2015.

Daellenbach, K. R., Bozzetti, C., Křepelová, A., Canonaco, F., Wolf, R., Zotter, P., Fermo, P., Crippa, M., Slowik, J. G., Sosedova, Y., Zhang, Y., Huang, R. J., Poulain, L., Szidat, S., Baltensperger, U., El Haddad, I., and Prévôt, A. S. H.: Characterization and source apportionment of organic aerosol using offline aerosol mass spectrometry, Atmos. Meas. Tech., 9, 23-39, 10.5194/amt-9-23-2016, 2016.

Hu, W. W., Campuzano-Jost, P., Palm, B. B., Day, D. A., Ortega, A. M., Hayes, P. L., Krechmer, J. E., Chen, Q., Kuwata, M., Liu, Y. J., de Sá, S. S., McKinney, K., Martin, S. T., Hu, M., Budisulistiorini, S. H., Riva, M., Surratt, J. D., St. Clair, J. M., Isaacman-Van Wertz, G., Yee, L. D., Goldstein, A. H., Carbone, S., Brito, J., Artaxo, P., de Gouw, J. A., Koss, A., Wisthaler, A., Mikoviny, T., Karl, T., Kaser, L., Jud, W., Hansel, A., Docherty, K. S., Alexander, M. L., Robinson, N. H., Coe, H., Allan, J. D., Canagaratna, M. R., Paulot, F., and Jimenez, J. L.: Characterization of a real-time tracer for isoprene epoxydiols-derived secondary organic aerosol (IEPOX-SOA) from aerosol mass spectrometer measurements, Atmos. Chem. Phys., 15, 11807-11833, 10.5194/acp-15-11807-2015, 2015.

McMeeking, G. R., Bart, M., Chazette, P., Haywood, J. M., Hopkins, J. R., McQuaid, J. B., Morgan, W. T., Raut, J. C., Ryder, C. L., Savage, N., Turnbull, K., and Coe, H.: Airborne measurements of trace gases and aerosols over the London metropolitan region, Atmos. Chem. Phys., 12, 5163-5187, 10.5194/acp-12-5163-2012, 2012.

Sciare, J., d'Argouges, O., Sarda-Estève, R., Gaimoz, C., Dolgorouky, C., Bonnaire, N., Favez, O., Bonsang, B., and Gros, V.: Large contribution of water-insoluble secondary organic aerosols in the region of Paris (France) during wintertime, Journal of Geophysical Research: Atmospheres, 116, n/a-n/a, 10.1029/2011JD015756, 2011.

Timonen, H., Saarikoski, S., Tolonen-Kivimäki, O., Aurela, M., Saarnio, K., Petäjä, T., Aalto, P. P., Kulmala, M., Pakkanen, T., and Hillamo, R.: Size distributions, sources and source areas of water-soluble organic carbon in urban background air, Atmos. Chem. Phys., 8, 5635-5647, 10.5194/acp-8-5635-2008, 2008.

Xu, L., Guo, H., Boyd, C. M., Klein, M., Bougiatioti, A., Cerully, K. M., Hite, J. R., Isaacman-VanWertz, G., Kreisberg, N. M., Knote, C., Olson, K., Koss, A., Goldstein, A. H., Hering, S. V., de Gouw, J., Baumann, K., Lee, S.-H., Nenes, A., Weber, R. J., and Ng, N. L.: Effects of anthropogenic emissions on aerosol formation from isoprene and monoterpenes in the southeastern United States, Proceedings of the National Academy of Sciences, 112, 37-42, 10.1073/pnas.1417609112, 2015.

Xu, L., Williams, L. R., Young, D. E., Allan, J. D., Coe, H., Massoli, P., Fortner, E., Chhabra, P., Herndon, S., Brooks, W. A., Jayne, J. T., Worsnop, D. R., Aiken, A. C., Liu, S., Gorkowski, K., Dubey, M. K., Fleming, Z. L., Visser, S., Prévôt, A. S. H., and Ng, N. L.: Wintertime aerosol chemical composition, volatility, and spatial variability in the greater London area, Atmos. Chem. Phys., 16, 1139-1160, 10.5194/acp-16-1139-2016, 2016.

---

## Author Comment (AC1) · 14 Sep 2016

**Author's response:**

We thank Referees #1 for the careful revision and comments which helped in improving the overall quality of the manuscript.

A point-by-point answer (in regular typeset) to the referees' remarks (in the *italic typeset*) follows, while changes to the manuscript are indicated in blue font.

In the following page and lines references refer to the manuscript version reviewed by anonymous referee #1

**Anonymous Referee #1**

*This manuscript presents an analysis of the composition and source apportionment of PM$_1$ filters collected at three sites in Lithuania. For this offline technique, the aqueous extracts from filters were nebulized with Ar for introduction into the HR-ToF-AMS. The use of Ar as the nebulization gas enabled an analysis of the CO$^+$/CO$_2^+$ fragment ratio and trends in that ratio with season. Positive matrix factorization was also applied on both the offline AMS data set as well as an offline marker data set collected using the same filters. This manuscript provides a good demonstration of the type of data sets that can be generated via this offline AMS technique and the CO+/CO2+ analysis provides new insights into the interpretation of AMS data from ambient samples. Thus, I see this paper as appropriate for publication in ACP. However, I have a few concerns, mostly related to sampling artifacts that need to be addressed prior to publication.*

1)      **P2 L9:** *Traffic exhaust OA is listed as a PMF factor from AMS spectra, yet in the experimental it is noted that the contribution is too low to be resolved with PMF and is instead estimated using a CMB approach. I suggest rewording the abstract to clarify this.*

We reworded the abstract as follows: "AMS WSOA spectra were analyzed using positive matrix factorization (PMF), which yielded 4 factors. These factors included biomass burning OA (BBOA), local OA (LOA) contributing significantly only in Vilnius, and two oxygenated OA (OOA) factors, summer OOA (S-OOA) and background OOA (B-OOA) distinguished by their seasonal variability. The contribution of traffic exhaust OA (TEOA) was not resolved by PMF due to both low concentrations and low water solubility. Therefore, the TEOA concentration was estimated using a chemical mass balance approach, based on the concentrations of hopanes, specific markers of traffic emissions."

**Changes in text:**

2)      **P5 L24:** *The nebulizer used was operated at 60°C, how long are the aerosols in this heated region? Was this temperature in the nebulizer also used in the Daellenbach et al. analysis? What effect might this high temperature have on the composition of the organics measured with the AMS compared to online analysis? If this temperature was not used for the Daellenbach analysis, what effect might this have on the factor specific recoveries of this work compared to the results from that previous analysis?*

The nebulizing Ar flow was 0.4 L min$^{-1}$. Considering the internal diameter (6 mm) and the length of our lines, we can estimate an aerosol residence time in our lines (from nebulization to AMS detection) of ca. 2 s. The aerosol residence time in the 60°C zone is significantly shorter (~100ms). A set of 40 PM$_1$ filter samples collected in Lithuania (not included within the source apportionment presented in this work) was measured using both the Apex Q nebulizer (Elemental Scientific Inc., Omaha NE 68131 USA) operated at 60°C and using a custom-built nebulizer (Daellenbach et al., 2016). The comparable WSOA/SO$_4^{2-}$ ratio registered using the two systems indicates a negligible loss of volatile organics (Fig. Discussion 1 (Fig. D1)).

We compared organic mass spectral time series and fragments fractional contributions retrieved from the two different nebulization systems. Mass spectra revealed a good correlation for all fragments ($R$ = 0.94 on average), similarly the total organic signal showed a correlation of $R$ = 0.94 (Fig. D1). Excluding CO$_2^+$ and the related fragments (CO$^+$, H$_2$O$^+$,

HO$^+$, and O$^+$, Aiken et al., 2008; Canagaratna et al., 2007), the intensity of which can be affected by the vaporizer history (Fröhlich et al., 2015, Pieber et al., 2016), we observed a good agreement between the normalized AMS mass spectral fingerprints obtained with the two different nebulizers, with 95% of the $i, j$ elements not statistically different within 2σ. As stated in the manuscript, here $i$, and $j$ represent a generic filter sample and a generic AMS fragment, respectively, while the uncertainty considered here includes blank variability, repeatability, uncertainty related to ion counting statistics and ion-to-ion signal variability at the detector. Overall the new nebulization system revealed a ~7 times higher sensitivity. Given the high correlation and the similarity in the mass spectral fingerprints, we can exclude substantial effects on the recoveries of the different factors.

[Figure]

Figure D1. Top: WSOA/SO$_4^{2-}$ ratio registered with a custom-made nebulizer (Daellenbach et al. 2016, here marked as "old nebulizer") and our nebulization system ("new nebulizer"). Bottom: OA signal comparison.

3) **P18 L25:** *PM$_1$ composition discussed here and shown in Figure 1 shows ions that can be measured with both the AMS and IC (e.g. SO4, NO3, etc.). Do the contributions shown in Figure 1 correspond to the IC measurements or AMS? For ions that can be quantified with both techniques, how do the values compare between the AMS and IC?*

**Author's response:**
As mentioned at P6, L30-31, the ion concentrations are from IC if not differently specified. For the sake of clarity we added this information in the Figure 1 caption.
Following the recommendations of anonymous referees #1 and #2 we added in the revised SI a comparison between offline-AMS and IC:

**Offline-AMS comparison with IC and WSOC determination by TOC analyzer**

Overall, the comparison between offline-AMS and IC concentrations of $NH_4^+$, $SO_4^{2-}$, and $NO_3^-$ reveals a non-linear relation due to the lower IC detection limits. This is most likely related to the low transmission efficiency of the AMS lens for small particles, particularly predominant for diluted filter extracts.

Nevertheless, considering internally mixed nebulized particles, the composition of the particles is not supposed to change with the solution concentration, as also confirmed by dilution tests conducted on our filter extracts (Fig. D2).

[Figure]

Figure D2. Dilution tests: NR PM composition and comparison of mass spectra registered at different dilutions.

[Figure]

Figure D3. Offline-AMS comparison with different techniques with IC and WSOC measurements by TOC analyzer.

Figure D2 and D3 were added to the SI as Fig. S16 and S17:
The following paragraph was added to Fig. S16 caption:

This low particle transmission efficiency for diluted solutions results in a high scattering at low concentration. Additional scattering is observed in the relation between offline-AMS and IC $SO_4^{2-}$. This is related to the presence of refractory sulfate salts (e.g. $Na_2SO_4$, ammonium sulfate) which are detectable by IC, but not with the AMS, consistent with lower slope obtained between offline-AMS and IC $SO_4^{2-}$, compared to the other species.

These species are likely formed during nebulization, e.g.
$$(NH4)_2SO_4 + CaCl_2 \rightleftharpoons CaSO_4 + 2NH_4Cl$$
For these reasons we only reported inorganic ion concentrations from IC.

4)    *P19 L14-20: The nitrate concentration shows clear seasonality with larger contributions in the winter and the sulfate concentration looks relatively constant*

*throughout the year. However, in Figure 1, the ammonium concentration appears to also be relatively constant throughout the year. Is this correct? If so, can the authors comment on potential counter ions for NO₃ ?*

**Author's response:**
Considering the $NH_4^+$, $SO_4^{2-}$ and $NO_3^-$ concentrations in µEq m$^{-3}$, the agreement between ($NH_4^+$) and ($SO_4^{2-}$ + $NO_3^-$) is high, with an average ($SO_4^{2-}$ + $NO_3^-$)/$NH_4^+$ ratio of 0.99 over the year and 1.02 during winter. The Pearson correlation coefficient $R$ between ($SO_4^{2-}$ + $NO_3^-$) and $NH_4^+$ was 0.92 considering the whole year and 0.84 considering only winter. Therefore, the role of other counter ions is negligible.

[Figure]

Figure D4. $NH_4^+$ correlation with $SO_4^{2-}$ + $NO_3^-$. Data in µEq m$^{-3}$ (top); ion balance (bottom).

Figure D4 was added to Fig. S11.

5)   **P20 L 28-31:** *The background-OOA factor appears to correlate with $NH_4^+$ much better at Preila and Vilnius than Rugsteliskes (Figure S11). Are there any potential reasons for the lower apparent correlation at Rugsteliskes? How much uncertainty is there in the $NH_4^+$ measurement? What is the significance of a correlation of B-OOA with $NH_4^+$?*

**Author's response:**
The B-OOA factor correlation with $NH_4^+$ is significant at all stations: $R = 0.82$ ($R^2 = 0.67$) for Vilnius, 0.87 ($R^2 = 0.76$) for Preila, and 0.71 ($R^2 = 0.50$) for Rūgšteliškis. The correlation of B-OOA with a secondary inorganic component such as $NH_4^+$ could suggest the secondary origin of B-OOA, as also inferred by the comparison with the marker-source apportionment (section 4.4.2). The repeatability of $NH_4^+$ IC measurements was 10%, while according to our error estimate (Section 3.1.3 ), the average relative uncertainty on the B-OOA factor for Rūgšteliškis was 12%. We estimated that up to half of the total unexplained variability in the relationship between $NH_4^+$ and B-OOA in Rūgšteliškis can be due to the abovementioned errors, while in Preila and Vilnius the B-OOA *vs* $NH_4^+$, most of the unexplained variability can be attributed to the errors. For Rūgšteliškis the remaining unexplained variability (27%) may be related to variability in the precursor composition and/or in the air masses photochemical age.
This information was added to Fig. S11 caption.

6) ***Section 2.1 and P21 L1-17:*** *Were the High-Volume samplers located in temperature controlled rooms? If not, what effect could higher summer temperatures have on the composition of the organic compared to the winter samples? Could the S-OOA factor be complicated by collection differences caused by the loss (on the filter) of more volatile organic molecules during summer months?*

**Author's response:**
High volume were equipped with temperature control systems maintaining the filter storage temperature always below 25°C, which is lower or comparable to the maximum daily temperature during summer (Fig. 3b). This should prevent large negative artifacts involving the most volatile fraction.

We added this information in P4, L16:

In order to prevent large negative filter artifacts, the high-volume samplers were equipped with temperature control systems maintaining the filter storage temperature always below 25°C, which is lower or comparable to the maximum daily temperature during summer.

7) ***P2 L6:*** *the $CO_2^+$:$CO^+$ ratios reported in section 4.5 are greater than 1. The less than sign should be switched.*

Corrected as suggested

8) ***P10 L22-23:*** *a verb such as "was used" is missing.*

Corrected as "was constrained"

9) ***P22 L3:*** *I suggest some mention directing the reader to Figure 5 be made in the text as the time series for the factors are discussed in this section but no mention of Figure 5 is made.*

We introduced a reference to Figure 5 at P22 L3

10) ***P25 L13:*** *"Using the ratio (1.88) calculated from offline-AMS". Suggest adding* $OM/OC_{BBOA}$ *ratio to communicate what ratio is being used in the calculation here.*

Corrected as suggested

11) ***P30 L 25-26:*** *suggest rephrasing, the double negative "unlikely return uncertain CO+ values" is confusing.*

Rephrased as: "should return accurate $CO^+$"

12) ***P45 Figure 2 and P46 Figure 4:*** *Suggest either writing out the factor names in the labels (background-OOA instead of B-OOA etc.) or giving the names and labels in the caption.*

Factor names and labels added in Figure 2 and Figure 4 captions.

---

## Author Comment (AC2) · 14 Sep 2016

**Author's response:**

We thank the Referees for the careful revision and comments which helped in improving the overall quality of the manuscript.

A point-by-point answer (in regular typeset) to the referees' remarks (in the *italic typeset*) follows, while changes to the manuscript are indicated in blue font.

In the following page and lines references refer to the manuscript version reviewed by anonymous referee #2.

**Anonymous Referee #2**

***General Comments:***

*This manuscript reported an analysis of $PM_1$ compositions and sources at three different sites in Lithuania based on filter samples. The authors applied AMS and other instruments to analyze the filter samples, and then performed PMF analysis to study the sources of OA and $PM_1$. This study presented a method/case to study the sources of total ambient OA based on the measurements of water soluble OA only. That is, apply PMF analysis on the water soluble organic mass spectra, identify multiple factors, and rescale the water soluble concentration to total concentration by applying recovery ratios. This is an interesting method but has large uncertainties, which arise from the recovery ratio. I think this manuscript is suitable for publication in ACP once the following comments have been addressed.*

***Source Apportionment***

We thank Anonymous Referee #2 for the careful review which indeed helped to improve the overall quality of our work. We want to state that while the uncertainty deriving from the recovery application is substantial, we do demonstrate that this uncertainty is comparable to that from PMF rotational uncertainty. The overall uncertainty of our source apportionment is factor dependent and is on average 14% for BBOA, 15% for B-OOA, 28% for S-OOA, and 100% for LOA, with the latter mostly due to the low concentrations during winter and . As a comparison, the $R_{BBOA}$ relative uncertainty ($\sigma R_{BBOA}$) was 10%, $\sigma R_{OOA}$ was 7%, and $\sigma R_{LOA}$ 14%. Our factor uncertainties are comparable to the AMS mass uncertainty, which is commonly considered to be 30%, but does not affect our results, and instead affects online-AMS source apportionment studies. Therefore the uncertainty relative to the offline-AMS methodology is high, yet comparable to the online-AMS source apportionment.

**Major comments**

1) *Ambient total OA source apportionment based on the measurement of water soluble OA.*

    *The major uncertainty of this method arises from the recovery ratio (Rz), which is a reflection of the bulk extraction efficiency and water solubility of OA factors. It is not clear how the Rz values are obtained in this study. As I understand, the authors randomly selected Rz from Daellenbach et al. (2016) as initial conditions and fit Eq. (6) to get RLOA. If so, how are RBBOA and ROOA obtained? Why are they different from the values in Daellenbach et al. (2016). Also, it is not clear which Rz values are eventually applied, from Daellenbach et al. (2016) or the values calculated in this study?*

As anonymous referee #2 mentioned, factor recoveries were randomly selected from the combinations reported in Daellenbach et al. (2016). The randomly selected $R_Z$ combinations were perturbed assuming possible biases in the OC and WSOC measurements in Daellenbach et al. (2016) and in this study. The perturbed randomly selected $R_Z$ combinations were then used as input to fit $R_{LOA}$ according to Eq. (6). Only $R_Z$ combinations leading to unbiased OC fit residuals were retained (i.e. OC fitting residuals not statistically different from 0 within 1σ for summer and winter individually and for the whole period). The retained $R_Z$ combinations were displayed as PDF in Fig. S8. The newly obtained $R_{BBOA}$ and

$R_{OOA}$ are systematically lower than those reported in Daellenbach et al. (2016), by 5.6% and 12.3% respectively, within the expected biases of the different measurements. L23 P12- L6, P13 were modified as follows:

For each of the 95 retained PMF solutions, Eq. (6) was fitted 100 times by randomly selecting a set of 100 $R_{BBOA}$, $R_{OOA}$ value combinations from those determined by Daellenbach et al. (2016). Each fit was initiated by perturbing the input $OC_i$ and $TEOC_i$ within their uncertainties, assuming a normal distribution of the errors. Additionally, in order to explore the effect of possible bulk extraction efficiency (WSOC/OC) systematic measurement biases on our $R_Z$ estimates, we also perturbed the OC, WSOC (Daellenbach et al., 2016) inputs. Specifically, we assumed an estimated accuracy bias of 5% for each of the perturbed parameters, which corresponds to the OC and WSOC measurement accuracy. In a similar way, we also perturbed the input $R_{BBOA}$ and $R_{OOA}$ assuming an accuracy estimate of 5% deriving from a possible OC measurement bias in Daellenbach et al. (2016) which could have affected the $R_Z$ determination. In total $9.5 \cdot 10^3$ fits were performed (Eq. 6) and we retained only solutions (and corresponding perturbed $R_Z$ combinations) associated with average OC residuals not statistically different from 0 within 1σ for each station individually and for summer and winter individually (~8% of the $9.5 \cdot 10^3$ fits, Fig. S6). The OC residuals of the accepted solutions did not manifest a clear correlation with the LOA concentration (Fig. S7), indicating that the estimated $R_{LOA}$ was properly fitted, without compensating for unexplained variability of the PMF model or biases from the other $R_z$. Fig. S8 shows the probability density functions (PDF) of the retained perturbed $R_z$ which account for all uncertainties and biases mentioned above.

2) *The authors mentioned that the bulk extraction efficiency in this study is lower than that in Daellenbach et al. (2016). This result is not surprising since one OA factor likely has contribution from multiple sources and the water solubility of OA factors may vary with site and season. For example, the water solubility of BBOA ranges from 64% to 80% (Sciare et al., 2011; Timonen et al., 2008). In addition, this method is not sensitive to primary OA factors (e.g., HOA and Cooking OA), which is largely water insoluble. This is another reason why HOA cannot be resolved from the PMF analysis. The limitations should be better discussed in the manuscript.*
*What suggestions do the authors have for researchers who want to use the method as proposed in this manuscript? For example, should they follow the same filter extraction procedures as in this study? How to calculate the Rz?*

Indeed, Bulk EE (WSOC/OC) can vary between site and seasons and WSOC ranges reported in the literature for the different sources (e.g. BBOA, *(Sciare et al., 2011; Timonen et al., 2008)* cover the ranges obtained here and in Daellenbach et al. (2016). However, it is unexpected that all primary and secondary factors determined in this study in both seasons have systematically lower water solubility than those in Daellenbach et al. (2016). By contrast, the Bulk EE differences found between this work and Daellenbach et al. (2016) can be fully explained by the WSOC and OC accuracy measurements.

The following recommendations for future offline-AMS users were added at P13 L19:

In general the recovery estimates reported in Daellenbach et al. (2016) represent the most accurate estimates available, being constrained to match the online-ACSM source apportionment results. The $R_Z$ combinations reported by Daellenbach et al. (2016) demonstrated to positively apply to this dataset, enabling properly fitting the measured Bulk EE (WSOC/OC) with unbiased residuals and therefore providing a further confidence on their applicability (we note that in Eq. 6 we fitted OC as function of $1/R_Z$ and $WSOC_{Z,i}$, therefore $R_Z$ fitted WSOC/OC = Bulk EE). In general further $R_Z$ determinations calculated

comparing offline-AMS and online-AMS source apportionments would be desirable in order to provide more robust $R_Z$ estimates. In absence of a-priori $R_Z$ values for specific factors (e.g. for LOA in this study) we recommend constraining the $R_Z$ combinations reported by Daellenbach et al. (2016) as a-priori information to fit the unknown recoveries, with the caveat that the $R_Z$ combinations reported by Deallenbach et al. (2016) were determined for filter samples water extracted following a specific procedure; therefore we recommend adopting these $R_Z$ combinations for filter samples extracted in the same conditions. Nevertheless the $R_Z$ combinations reported by Daellenbach et al. (2016) should be tested also for filters extracted with water in different conditions to verify whether they can properly fit the Bulk EE. In case the $R_Z$ combinations reported by Daellenbach et al. (2016) would not apply for a specific location or extraction procedure (i.e. not enabling a proper fit of Bulk EE) we recommend a $R_Z$ redetermination by comparing the offline-AMS source apportionment results with well-established source apportionment techniques. In absence of data to perform a well-established source apportionment, we recommend to fit all the $R_Z$ to match the bulk EE (i.e. fitting all the recoveries similarly as in Eq. 6 without constraining any a-priory $R_Z$ value).

In general, the offline-AMS technique assesses less precisely the contribution of the lower water soluble factors. The higher uncertainty mostly stems from the larger PMF rotational ambiguity when separating a factor characterized by low concentration in the aqueous filter extracts. Nevertheless, the uncertainty is dataset dependent, as the separation of source components with low water solubility can be improved in case of distinct time variability characterizing those sources in comparison with the other aerosol sources. The low aqueous concentration of scarcely water soluble sources in fact can be partially overcome by the large signal/noise characterizing the offline-AMS technique (170 on average for this dataset).

> *3)      Discussions on instruments comparison are required.*
>
> *Inorganic ions such as NH4+, NO3-, and SO42- are measured by both AMS and IC. The authors should present the instruments comparison.*

The comparison between offline-AMS and IC ion concentrations was discussed and added to the SI, according also to Anonymous Referee #1 question (question 5). We note though that offline AMS data are not used for quantification, which will be the subject of an up-coming study.

> *4)      Page 9 Line 29-30. The AMS measured concentration is scaled to match the WSOC measurement. What's the scale ratio? Is the scale ratio the same for all filter samples?*

Similarly to $NH_4^+$, $SO_4^{2-}$, and $NO_3$, and for the same reasons discussed above (Anonymous referee #1, question 5), the WSOC signal from offline-AMS does not follow a linear relation. Therefore the scaling factor is not constant. We would like to note once again that the AMS has not been used for quantification, specifically because of these issues related to particle transmission efficiency; moreover, as displayed in Fig. D2 the WSOM AMS mass spectral fingerprint does not show large changes when diluting our filter extracts. This comparison was inserted in the revised SI.

[Figure]

Figure D2.. Dilution tests: NR PM composition and comparison of mass spectra registered at different dilutions.

[Figure]

Figure D5. Correlation between WSOC offline-AMS signal and WSOC measurements by TOC analyzer.

5)        *The difference in separation and classification of OA factors between online and offline-AMS (Page 20 Line 14-27).*

*I disagree with the statement that "online-AMS OOA factors are commonly classified based on their volatility", because chemistry and sources also affect the factor separation. For example, the separation of IEPOX-OA factor (Budisulistiorini et al., 2013; Hu et al., 2015) or called isoprene-OA factor (Xu et al., 2015) is driven by IEPOX chemistry, but not volatility. Also, Xu et al. (2015) showed that nighttime monoterpene oxidation by nitrate radical contributes to less-oxidized OOA (as termed SV-OOA in this study).*

Following the suggestion of anonymous referee #2 we modified the lines at P20 L17-18 as follows:

Few online-AMS studies reported the separation of isoprene-related OA factor (Budisulistiorini et al., 2013; Hu et al., 2015, Xu et al., 2015) mostly driven by isoprene epoxides chemistry. Xu et al. (2015) showed that nighttime monoterpene oxidation by nitrate radical contributes to less-oxidized OOA. However, the large majority of online-AMS OOA factors are commonly classified based on their volatility (semi-volatile OOA and low-volatility OOA) rather than on their sources and formation mechanisms.

6)        *The authors stated that "the offline-AMS sources apportionment separates factors by seasonal trends rather than volatility". However, sometimes, seasonal trend affects the source apportionment through volatility. For example, Page 23 Line 26-27 discussed that higher NO3--related SA exhibits higher concentration in winter than summer, which is due to the semi-volatile nature of NO3- (Page 19 Line 20).*

Concerning the relation between seasonality and volatility, we agree that OOA factors with different seasonal behaviors can be characterized by different volatilities. However in this work the offline-AMS OOA separation is not driven by volatility, given the low correlation between $NO_3^-$ and our OOA factors (this is also reflected by the low $NO_3^-$-related SOA correlation with B-OOA and S-OOA, Table 2). Additionally, the partitioning of semi-volatile OA at low temperatures would lead to a less oxidized OOA fingerprint during winter; however, this is not the case here. We observed a less oxidized OOA factor during summer,

whose fingerprint closely resembles that of SOA from biogenic precursors, while similar to OOA from biomass burning emissions OOA during the cold season is more oxidized. This has been also reported from online-ACSM monitoring campaigns (Canonaco et al., 2015),

7) *OM/OC ratio.*

*In this study, the OM/OC is calculated by Aiken method (Page 12 Line 20). However, a recent study by Canagaratna et al. (2015) improved the estimation from Aiken method by including composition-dependent correction factors. The Canagaratna method is recommended to use. Since many calculations in this study depend on the OM/OC ratio, how would it affect the results/conclusions if the authors use Canagaratna method to calculate the OM/OC ratio?*

Following the suggestion of anonymous referee #2 we included following discussion within the SI.

We recalculated the OM:OC ratio for the water soluble collected spectra according to the new parametrization reported by Canagaratna et al. (2015). Consistently with Canagaratna et al. (2015), the newly calculated OM:OC ratio was on average 9% higher than the OM:OC ratio calculated according to Aiken method. More specifically, the OM:OC ratio was on average 9% higher during summer, and 10% during winter. The two methods reported well correlated OM:OC values ($R = 0.98$ over the whole monitoring period, $R = 0.99$ during winter, $R = 0.97$ during summer). In our study, the OM:OC ratios of our water soluble mass spectra were mostly used to determine the total WSOM concentrations. Considering the high correlations between the Aiken and Canagaratna OM:OC ratios, we can exclude large effects on the WSOM variability and therefore on the source apportionment. Nevertheless the WSOM estimated concentrations would be 10 % larger, when assuming the Canagaratna OM:OC parametrization. In general Aiken assumed a $CO_2^+:CO^+$ ratio of 1, while Canagaratna stated that such an assumption would underestimate $CO^+$. From our dataset, we observed a $CO_2^+:CO^+$ of $1.75_{med}$ suggesting that the Aiken OM:OC parametrization would represent more accurately our data although both parametrizations are uncertain for this dataset.

8) *Background-OOA (B-OOA) factor.*

*When the authors selected solutions, one criterion is the correlation between B-OOA and NH4+ (Page 12 Line 8). The authors should explain the use of NH4+. SO42- is regional and usually used as background OA. What's the correlation between B-OOA and SO42-? In Page 20 Line 30, it is stated that B-OOA correlates well with NH4+. However, the correlation between B-OOA and NH4+ varies with site as shown in Fig. S11. For example, the correlation is really weak for the Rugsteliskis site.*

The lower correlation between $NH_4^+$ and B-OOA in Rūgšteliškis ($R^2 = 0.5$ vs $R^2 > 0.7$ at other locations) and its possible explanation were discussed in the response to anonymous referee #1 (question 5). The repeatability of $NH_4^+$ measurements is estimated to be around 10%, while according to our error estimate (Section 3.1.3), the average relative uncertainty on the B-OOA factor for Rūgšteliškis was 12%. We estimated that up to half of the total unexplained variability in the relationship between $NH_4^+$ and B-OOA in Rūgšteliškis can be due to the abovementioned errors, while for the B-OOA *vs* $NH_4^+$ relationship in Preila and Vilnius most of the unexplained variability can be attributed to these errors. For Rūgšteliškis the remaining unexplained variability (27%) can be related to variability in the secondary precursor composition and/or in the air masses photochemical age.

The criterion based on the $NH_4^+$ *vs* B-OOA correlation did not reveal any negative correlation for each station individually and for all the stations together, therefore no PMF solution was discarded according to this criterion as well as for the criterion based on the correlation of levoglucosan with BBOA (this information was added to the manuscript). As previously discussed, $NH_4^+$ [$\mu Eq\ m^{-3}$] matches the sum of $SO_4^{2-}$ and $NO_3^-$ [$\mu Eq\ m^{-3}$]. Therefore $NH_4^+$ variability well represents the variability of inorganic secondary components of different origin (local: $NO_3^-$ and regional: $SO_4^{2-}$) formed at different time scales. Nevertheless, similar to B-OOA retrieved from the offline-AMS PMF, $NH_4^+$ correlates most significantly with sulfate ($R$ = 0.80) and the sulfate-rich factor from the marker-PMF, indicating that these species represent the background long range transported aerosols.

9) *If B-OOA represents background OA, why is B-OOA lower in urban site than the other sites? I disagree with the authors' argument that this difference is caused by PMF residual uncertainties or biases (Page 29 Line 10). The authors' argument is flawed because it is based on circular assumptions. When the authors calculate B-OOAmarker, the LOA and S-OOA are based on PMF analysis without considering "some residual uncertainties or biases". If the authors considered "some residual uncertainties or biases" and re-performed PMF analysis, the concentrations of LOA and S-OOA would change, which would influence and concentration of B-OOAmarker. In that circumstance, B-OOAoffline-AMS may agree among all three sites, but B-OOAmarker may be different among all three sites.*

Showing that PMF results are affected by model residuals is exactly the point we wanted to make with this comparison. Therefore, drawing strong conclusions on site-to-site differences should be done with caution. In the current version of the manuscript we elaborate further on these issues, as we discuss below. The discussion regarding B-OOA differences at different sites was modified as follows (added in P26, L31):

Another advantage obtained in coupling the two source apportionment results is the possibility to study the robustness of the factor analyses by evaluating the consistency of the two approaches as we already discussed for the primary OA and Other-OA fractions. Figure S14a displays the PMF modelled WSOC:measured WSOC PMF for the offline-AMS case, indicating a clear bias between Vilnius and the rural sites, with a WSOC overestimation of ~5% in Preila and Rūgšteliškis. While this overestimation is negligible for WSOC mass, it might have significant consequences on single factor concentrations. By contrast, OM residuals are more homogeneous for the case of markers PMF (Fig. S14b). As we show in Fig. S6, these residuals marginally affect the apportionment of combustion sources, as suggested by the well comparing estimates of BBOA and TEOA using the two methods. Therefore, these residuals are more likely affecting non-combustion sources (LOA, S-OOA and B-OOA). For the common days, the S-OOA concentration is not statistically different at the different stations during summer (confidence interval of 95%), indicating that the residuals are more likely affecting LOA and B-OOA, which instead show site-to-site differences. Now, the PMF WSOC residuals appear at all seasons, also during periods without significant LOA contribution in Vilnius. Therefore, we conclude that B-OOA is the factor most significantly affected by the difference in the WSOC residuals. We could best assess the residual effects by comparing the B-OOA_offline-AMS with that estimated using the other technique that seem to yield more homogeneous residuals: B-OOA_marker. Here B-OOA_marker is estimated as Other-OA_markers - LOA - S-OOA. While B-OOA_offline-AMS shows site-to-site differences, B-OOA_markers did not show statistically different concentrations at all

stations within a confidence interval of 95%. Based on these observations, we conclude that observed site-to-site differences in B-OOA concentrations are likely to be related to model uncertainties.

[Figure]

Figure D6. a) Modelled OM : input OM for the markers-PMF. b) Modelled WSOC : measured WSOC for the offline-AMS PMF

Figure D6 was added to revised SI as Fig. S14

*Minor comments*
10)     *TEOA is resolved from CMB, not PMF. This needs to be clarified in multiple places in the manuscript, such as Page 2 Line 9 and Page 23 Line 30. Considering that the TEOA concentration is small and only one filter has statistical significant TEOA concentration (Page 22 Line 27), I suggest the authors to remove the comparison about TEOA concentration between sites (for example, Page 32 Line 15-17).*

We clarified in P2, L9, P25, L14, and P 23 L30 that PMF returned 4 factors, and TEOA was estimated by CMB. We replaced the TEOA comparison between sites with the comparison of the hopanes concentration at the different locations (P 25 L19, P26, L31-32, and P32 L 15-17).

> 11) *Page 2 Line 10. Please rephrase to "two oxygenated OA factors, summer OOA (S-OOA) and background OOA (B-OOA)".*

Corrected as suggested.

> 12) *Page 2 Line 16 vs. Line 18. Use OA or OM. Be consistent.*

Corrected as suggested.

> 13) *Page 4 Line 3. Please rephrase to "source apportionment on the submicron water soluble OA" in order to be precise about the method.*

We agree with anonymous referee #2 that our method access only the water soluble fraction, however the water soluble factor concentrations obtained from PMF analysis were subsequently rescaled for the corresponding factor recoveries enabling accessing the total OA concentrations (as also previously pointed out by anonymous referee #2, the recovery correction increases the uncertainty of our source apportionment).

> 14) *Page 5 Line 24. The nebulizer temperature is 60°C, which is different from Daellenbach et al. (2016). Also, the nebulizer system in this study is different from that in Daellenbach et al. (2016). Would these differences cause the difference in Rz between studies?*

As previously discussed (anonymous referee #1, question 2), the use of two different nebulizing setups are unlikely to significantly affect our source apportionment results and therefore our *Rz* estimates. This is due to the well comparing time series of fragments and mass spectral fingerprints. The differences in the *Rz* estimates stem from the different bulk EE (WSOC/OC) values measured for the two different datasets. We note that those differences can be fully ascribed to WSOC and/or OC measurement biases assuming a mass accuracy of 5% for both measurements.

> 15) *Page 5 Line 27-28. The correction of blank is not appropriate. This is because the particles generated from nebulizing DI water only are too small to be detected by AMS. However, the organics associated with DI water will be detected by AMS when nebulizing real filter extracts because the particles are big. I suggest the authors to nebulize ammonium sulfate solution (i.e. dissolve ammonium sulfate in DI water with similar concentration as ambient filters) and use the detected organic concentration as blank.*

In this study we nebulized twice per day a $NH_4NO_3$ solution. We compared our blank OA mass spectra with the OA mass spectra collected during $NH_4NO_3$ nebulization. Excluding $CO_2^+$ and the related fragments, which can be affected by $NH_4NO_3$ induced non-OA $CO_2^+$ signal, (Pieber et al. 2016, Friedel et al., 1953, Friedel et al., 1959), none of the other OA AMS fragments showed significantly different concentration from our blanks (ultrapure water nebulization) within 2σ. Our average signal to blank ratio was 170, indicating that the blank represented only a small fraction of the total signal. . Therefore, we consider that under our conditions the nebulization of pure water and $NH_4NO_3$ solution yield equivalent results. Nevertheless, we recognize that nebulizing $(NH_4)_2SO_4$ or $NH_4NO_3$ solutions would provide a better estimate of the OA blank. This methodology can be indeed implemented for future studies.

 *Page 9 Line 7-9. Although the detailed procedures have been discussed in Daellenbach et al. (2016), it is still helpful to briefly discuss the method in the manuscript, especially how the recovery ratios are calculated.*

Rephrased as: "The offline-AMS source apportionment returns the water soluble PMF factor concentrations. Daellenbach et al. (2016) determined factor specific recoveries (including PMF factor extraction efficiencies), by comparing offline-AMS and online-ACSM OA source apportionments. In particular, the filter samples were collected for one year during an online-ACSM monitoring campaign conducted at the same sampling station. Briefly, the factor recoveries were determined as the ratio between the water soluble OA PMF-factor concentrations retrieved from offline-AMS source apportionment divided by the OA PMF factor concentrations obtained from ACSM OA source apportionment. Factor specific recoveries and corresponding uncertainties were determined for HOA, BBOA, COA, and OOA".

17) *Page 10 Line 28. Please rephrase to "this factor has too small contribution in the water extracts to be resolved".*

Corrected as suggested.

18) *Page 12 Line 6. This sentence has been repeated twice. Delete.*

Sentence deleted as suggested

19) *Page 12 Line 13-16. AMS measures OM, instead OC. Please be clear that the conversion from OM to OC is for the carbon mass closure in Eq. (6).*

The information was added to the manuscript as suggested: "Here the water-soluble OA factor concentrations were converted to the corresponding water-soluble OC concentrations to fit the measured OC."

20) *Page 12 Eq. (6). WSW-OOA should be WSB-OOA. Is Rz the same for S-OOA and B-OOA since the same ROOA is applied for both factors?*

WSW-OOA was corrected as WSB-OOA

In this study we assumed $R_{S-OOA} = R_{B-OOA}$ because the recoveries of the OOA factors reported in Daellenbach et al. (2015), were determined from the sum of two OOA factors. The two recoveries were not determined individually in Daellenbach et al. (2015) due to the dissimilar OOA classification between offline-AMS and online ACSM source apportionments, which prevented an unambiguous attribution of the offline-AMS OOA factors to the online-AMS ones.

21) *Page 14 Line 20. What's the OMres/OM ratio?*

The information was added to the manuscript: "$OM_{res}$ represented on average 95±2% of total OM."

22) *Page 15 Line 21. List the non-source specific variables.*

The information was added to the text: "(EC, $OM_{res}$, (Me-)PAHs, S-PAHs, inorganic ions, oxalate, alkanes)".

The entire list is reported here below:

(EC, $SO_4^{2-}$, $NO_3^-$, $Cl^-$, $NH_4^+$, $Na^+$, $K^+$, $Ca^{2+}$, $Mg^{2+}$, oxalate, MSA, Phenanthrene, anthracene, fluoranthene, pyrene, benzo[a]anthracene, chrysene, triphenylene, retene, benzo[b,k]fluoranthene, benzo[j]fluoranthene, benzo-e-pyrene, benzo[a]pyrene, indeno[1,2,3

- cd]pyrene, dibenzo[a,h]anthracene, benzo[ghi]perylene, coronene, dibenzothiophene, phenanthro(4,5-bcd)thiophene, Benzo(b)naphtho(2,1-d)thiophene, Benzo(b)naphtha(1,2-d)thiophene, Benzo(b)naphtho(2,3-d)thiophene, Dinaphtho(2,1-b;1',2'-d)thiophene, Benzo(b)phenantho(2,1-d)thiophene, 2-methylnaphtalene, 1-methylfluoranthene, 3-methylphenanthrene, 2-methylphenanthrene, 2-methylanthracene, 4/9 methylphenanthrene, 1-methylphenanthrene, 4-methylpyrene, 1-methylpyrene, 1+3-methylfluoranthene, methylfluoranthene/pyrene, 3-methylchrysene, methylchrysene/benzoanthracene, Cholesterol, 6,10,14-trimethyl-2-pentadecanone, Undecane (C11), dodecane (C12), tridecane (C13), tetradecane (C14), pentadecane (C15), exadecane (C16), heptadecane (C17), octadecane (C18), nonadecane (C19), eicosane (C20), heneicosane (C21), docosane (C22), tricosane (C23), tetracosane (C24), pentacosane (C25), hexacosane (C26), heptacosane (C27), octacosane (C28), nonacosane (C29), triacontane (C30), untricontane (C31), totriacontane (C32), tritriacontane (C33), tetratriacontane (C34), pentatriacontane (C35), hexatriacontane (C36), heptatriacontane (C37), octatriacontane (C38), nonatriacontane (C39), tetracontane (C40), pristane, phytane, $OM_{res}$)

23) *What's the Hopanes_sum/OC ratio in the traffic exhaust factor? Is it consistent with the CMB method (i.e., 0.0012 in Page 11 Line 15)?*

Since our HOA matches between the two methods within our uncertainty, also the Hopanes_sum:OC ratio will be not statistically different. Note that the hopanes were constrained to contribute only to traffic in the markers source apportionment (Section 5.3.2.2).

24) *Page 16 Line 25. Should be "EC/OMres" ratio.*

Text corrected as "while EC:BB ratio was constrained to 0.1".

25) *Page 17 Line 10-16. The discussion is not clear. Suggest re-wording.*

Lines 10-16 were reformulated as:

As discussed in section 3.2.2, we assumed the contribution of specific markers to be 0 in different factor profiles. Such assumptions preclude the PMF model to vary the contributions of these variables from 0 (Eq. 3). In order to explore the effect of such assumptions on our PMF results we loosened all these constraints assuming variable contributions equal to 50%, 37.5%, 25%, and 12.5% of their average relative contribution to measured $PM_1$. In all cases the *a*-value was set to 1.

26) *Page 20 Line 1-3. List the levoglucosan/BBOC range in the literature. Similar suggestions for other places. For example, list the non-fossil primary organic carbon in Page 25 Line 13 and average fossil primary OC in Page 25 Line 29.*

Information added to the manuscript.

27) *Page 21 Line 2. I disagree with that S-OOA increases exponentially with average daily temperature from the data points in this study (Fig. S12). For example, many data points with T > 25°C do not have high S-OOA concentration and do not follow the exponential fit.*

Indeed data show a certain scattering. This scattering can stem from other parameters affecting the biogenic SOA concentrations, such as the photochemical aging of the air parcel, RH, rain, solar radiation, $NO_x$ concentration, accumulation during the previous days, and wind speed. When binning the data from Lithuania and Payerne in temperature steps of

5 degrees the exponential relation of S-OOA *vs* average daily temperature reveals a good agreement with the exponential relation reported by Leaitch et al. (2011). We also modified Fig. S12 adding the error bars and binning the S-OOA concentration in 5°C temperature steps.

[Figure]

Figure D7. S-OOA temperature dependence and submicron forest organic aerosol mass (SFOM) temperature parameterization by Leaitch et al. (2015). a) Lithuania; b) rural site of Payerne (Switzerland), Bozzetti et al. (2016); c) Binned S-OOA concentrations (average and standard deviation).

28) *Page 22 Line 13-15. This has been mentioned previously in Page 20 Line 1-3. It is not proper to discuss BBOC here because this section focuses on the marker-PMF, instead of offline AMS. Similar problem for Page 22 Line 23-24.*

The levoglucosan:BBOC ratios discussed in this section (P22 L13-15 and 23-24) actually refer to the marker-PMF source apportionment. In order to estimate the BBOC concentration from the marker source apportionment we used the OM:OC$_{BBOA}$ ratio retrieved from offline-AMS.

29) *Page 23 Line 14-15. The observation that nitrate concentration is higher in urban site than rural site has been shown in many previous studies (Xu et al., 2016; McMeeking et al., 2012), which should be cited here.*

Citations added as suggested

30) *Page 23 Line 30-31. This sentence is confusing. The remaining OM fraction is termed as OMres in Page 10 Line 20, but termed as Other-OA here. It should be clearly stated that Other-OA refers to OA after excluding BB and TE.*

Text corrected as suggested: "(Other-OA = OA – BB - TE)"

31) *Page 24 Line 18. Should be "higher"*

Text corrected as suggested

32) *Page 24 Line 21-23. (1) Which method did the authors use to get the BBOA concentration and correlation in this sentence? (2) It would be helpful to include a scatter plot between Preila and Vilnius. (3) I disagree with "the importance of regional meteorological conditions" as stated in this sentence and Page 32 Line 31-32. Firstly, the BBOA concentrations are different between two sites. Secondly, the BBOA in the Rugsteliskis site does not correlate with the other two sites.*

(1) The BBOA concentration reported at P24 L21-23 was estimated by offline-AMS. Information added to the text.

(2-3) For this comparison we considered only filter samples collected simultaneously during winter at the different stations. In this case we observed high correlations between the winter BBOA concentrations estimated for Preila and Vilnius ($R = 0.91$), and significantly positive correlations between Preila and Rūgšteliškis ($R = 0.72$) and between Vilnius and Rūgšteliškis ($R = 0.66$). We do not mean that BBOA has a regional origin, as also confirmed by the different concentrations observed at the different stations. The high correlations between the sites only suggest either a common accumulation/depletion of pollutants due to similar meteorological conditions, or a concomitant increment/decrease of residential wood combustion activity at the different stations. We could exclude the latter hypothesis because, as mentioned in the text, most of the BBOA spikes were not directly related to a decrease of temperature (Section 4.4.1)y. Therefore the BBOA daily variability in the region seem to be mostly driven by regional meteorological patterns (rain episodes and anticyclonic conditions), however, the proximity to biomass burning emission spots can influence the total concentration, therefore not surprisingly Vilnius and Preila show higher concentrations than Rūgšteliškis.

[Figure]

Figure D8. S-OOA temperature dependence and submicron forest organic aerosol mass (SFOM) temperature parameterization by Leaitch et al. (2015). a) Lithuania; b) rural site of Payerne (Switzerland), Bozzetti et al. (2016); c) Binned S-OOA concentrations (average and standard deviation).

P24 Lines 21-23 were corrected as:

During winter, considering only the samples collected concomitantly, Preila and Vilnius showed well correlated BBOA time series ($R$ = 0.91) and significantly positive correlations were observed for also for Preila and Rūgšteliškis ($R$ = 0.72) and for Vilnius and Rūgšteliškis ($R$ = 0.66) (offline-AMS BBOA time series). These results highlight the effect of regional meteorological conditions on the BBOA daily variability in the south east Baltic region.

*33)      Page 24 line 29. Both methods have the same time resolution (one filter per day).*

As mentioned in the main text in Table 1, Table S1, section 2.3 and section 3.2.1 this is not the case as the marker-source apportionment is based on composite samples which were created by merging two consecutively collected filter samples, and therefore the time resolution is 48 h.

*34)      Page 25 line 15. In the statistical significance test, why is sometimes 1σ is used but sometimes 3σ is used (for example, Page 26 Line 28).*

We homogenized all the statistical significances to the confidence interval of 3σ.

*35)      Page 26 Line 30. Should be "factor" instead of "fraction".*

Corrected as suggested

*36)      Table 2. The correlation coefficient R between NO3-related SOA and B-OOA is only 0.21. Thus, it is not meaningful to discuss the relationship between NO3-related SA and B-OOA (Page 28 Line 17). Similar problem for the relationship between MSA-related SOA and S-OOA (Page 28 Line 21).*

The $NO_3$-related SOA correlation with B-OOA is indeed small, however the correlation with LOA and S-OOA is negative, suggesting that the mass attributed to $NO_3$-related SOA by the markers source apportionment is fully attributed to the B-OOA factor in the offline-AMS source apportionment. This is also confirmed by the fact that the sum of LOA and S-OOA concentrations during winter (when the $NO_3$-related SOA substantially contributes) can't explain the $NO_3^-$-related SOA mass, which therefore has to be attributed to B-OOA. We believe that this result is relevant because it relates the $NO_3^-$-related SOA factor, typically resolved from a marker source apportionment, to the OOA factor typically resolved by AMS source apportionment in winter datasets. In a similar way we found that large part of MSA-related SOA is related to S-OOA, which provides more insight into the S-OOA precursors, moreover the precursor emissions of both factors (dimethyl sulfide, isoprene, and terpenes) are known to be strongly related to temperature, and not surprisingly the two factors increase during summer.

Lines 17-20, P28 were modified as follows:

The $NO_3^-$-related SOA and the PBOA were mostly related to the B-OOA factor as they showed higher correlations with B-OOA than with S-OOA. The B-OOA factor therefore may explain a small fraction of primary sources (PBOA), which however represents only 0.6%$_{avg}$ of the total OA. In detail, the $NO_3$-related SOA correlation with B-OOA was poor ($R$ = 0.21), however the correlation with LOA and S-OOA was negative (Table 2), suggesting that the mass attributed to $NO_3$-related SOA by the markers source apportionment was fully attributed to the B-OOA factor in the offline-AMS source apportionment. This is also confirmed by the fact that the sum of LOA and S-OOA concentrations during winter (when the $NO_3$-related SOA substantially contributes) can't explain the $NO_3^-$-related SOA mass, which therefore has to be attributed to B-OOA.

We added the following discussion at P 28, L26.

The correlation between the two factors is therefore not surprising as the precursor emissions (dimethyl sulfide, isoprene and terpenes) are strongly related to temperature leading to higher summer MSA-related SOA and S-OOA concentrations.

37)    *Page 29 Line 18. Please rephrase to "fCO2 value is higher than fCO".*

Corrected as suggested

38)    *Page 29 Line 24-25. The logic is not clear. Why does higher CO2+/CO+ ratio of gas CO2 suggest a minor contribution from WSOM decarboxylation to CO+.*

L24-25, P29 were modified as follows:

The fragmentation of pure gaseous $CO_2$ returned a $CO_2^+:CO^+$ ratio of $8.21_{avg}$ which is significantly higher than our findings for the water-soluble bulk OA ($1.75_{med}$). Assuming thermal decarboxylation of organic acids as the only source of $CO_2^+$ does not explain the observed $CO_2^+:CO^+$ ratio of $1.75_{med}$ and another large source of $CO^+$ has to be assumed. Therefore, the carboxilic acid decarboxylation into $CO_2$ can be considered as a minor source of $CO^+$.

39)    *Page 30 Line 7. Many data points from the Rugsteliskis site are outside the triangle range in Fig. 7a.*

As discussed in Fig. 7 caption, some points from Rūgšteliškis lie outside the triangle, suggesting that $CO^+$ and $CO_2^+$ variabilities are not well explained by our PMF model for those specific filter samples. However, Fig. S5 displays flat residuals for Rūgšteliškis, indicating an overall good WSOM explained variability by the model.

40)    *Page 31 Line 4. The correlation between CO+ and C2H3O+ is not shown in Fig. 7b. It would be helpful to show a scatter plot.*

We added to Fig. 7 the scatter plot $fCO^+$ vs. $fC_2H_3O^+$ as suggested.

41)    *Page 31 Line 16. Canagaratna et al. (2015) carefully discussed the CO2+/CO+ ratio of a number of standards, which should be discussed and mentioned more in the manuscript.*

As mentioned in the manuscript (P31, L24), we can observe that the most representative standards of our aqueous filter extracts in terms of $CO^+:CO_2^+$ ratio were multifunctional carboxylic acids (only hydroxyl mono and poly-acids and keto acids) and 2 diacids used by Canagaratna et al. (2015) . Specifically, These include citric acid, malic acid tartaric acid, ketobutyric acid, hydroxyl methylglutaric acid, pyruvic acid, oxaloacetic acid, tartaric acid, oxalic acid and malonic acid. Considering that the median OA bulk extraction efficiency was 0.59, and considering that the $CO^+$ and $CO_2^+$ fragmentation precursors tend to be more water soluble than the bulk OA, the listed compounds could be representative of large part of the $CO^+$ and $CO_2^+$ fragmentation precursors.

Lines 23-28, P31 were modified as follows:

With the exception of some multifunctional compounds (citric acid, malic acid tartaric acid, ketobutyric acid, hydroxyl methylglutaric acid, pyruvic acid, oxaloacetic acid, tartaric acid, oxalic acid and malonic acid), the water-soluble single compounds analyzed by Canagaratna et al. (2015) mostly showed $CO_2^+:CO^+$ ratios <1, systematically lower than the $CO_2^+:CO^+$ ratios measured for the bulk WSOM in Lithuania (1$^{st}$ quartile 1.50, median 1.75, 3$^{rd}$ quartile 2.01), which represents a large fraction of the total OM (bulk EE: median = 0.59, 1$^{st}$ quartile = 0.51, 3$^{rd}$ quartile = 0.72). Considering the relatively high extraction efficiency, and

considering that the $CO^+$ and $CO_2^+$ fragmentation precursors tend to be more water soluble than the bulk OA, the aforementioned compounds could be representative of a large part of the $CO^+$ and $CO_2^+$ fragmentation precursors.

*42)   Figure 5. The grey caps of traffic exhaust are not clear in this figure.*

Traffic grey caps were highlighted with a marker

[Figure]

Figure D9. Figure 5. $PM_1$ marker source apportionment: factor time series and relative contributions. Shaded areas indicate uncertainties (standard deviation) of 20 bootstrap runs.

References:

Budisulistiorini, S. H., Canagaratna, M. R., Croteau, P. L., Marth, W. J., Baumann, K., Edgerton, E. S., Shaw, S. L., Knipping, E. M., Worsnop, D. R., Jayne, J. T., Gold, A., and Surratt, J. D.: Real-Time Continuous Characterization of Secondary Organic Aerosol Derived from Isoprene Epoxydiols in Downtown Atlanta, Georgia, Using the Aerodyne Aerosol Chemical Speciation Monitor, Environ Sci Technol, 47, 5686-5694, Doi 10.1021/Es400023n, 2013.

Friedel, R. A., Sharkey, A. G., Shultz, J. L., and Humbert, C. R.: Mass spectrometric analysis of mixtures containing nitrogen dioxide, Anal. Chem., 25, 1314-1320, 1953.

Friedel, R. A., Shultz, J. L., and Sharkey Jr, A. G.: Mass spectrum of nitric acid, Anal. Chem., 31, 1128, 1959.

Fröhlich, R., Crenn, V., Setyan, A., Belis, C. A., Canonaco, F., Favez, O., Riffault, V., Slowik, J. G., Aas, W., Aijälä, M., Alastuey, A., Artiñano, B., Bonnaire, N., **Bozzetti, C.**, Bressi, M., Carbone, C., Coz, E., Croteau, P. L., Cubison, M. J., Esser-Gietl, J. K., Green, D. C., Gros, V., Heikkinen, L., Herrmann, H., Jayne, J. T., Lunder, C. R., Minguillón, M. C., Mocnik, G., O'Dowd, C. D., Ovadnevaite, J., Petralia, E., Poulain, L.,

Priestman, M., Ripoll, A., Sarda-Estève, R., Wiedensohler, A., Baltensperger, U., Sciare, J., and Prévôt, A. S. H.: ACTRIS ACSM intercomparison – Part 2: Intercomparison of ME-2 organic source apportionment results from 15 individual, co-located aerosol mass spectrometers, Atmos. Meas. Tech., 8, 2555–2576, 2015.

Hu, W. W., Campuzano-Jost, P., Palm, B. B., Day, D. A., Ortega, A. M., Hayes, P. L., Krechmer, J. E., Chen, Q., Kuwata, M., Liu, Y. J., de Sá, S. S., McKinney, K., Martin, S. T., Hu, M., Budisulistiorini, S. H., Riva, M., Surratt, J. D., St. Clair, J. M., Isaacman-Van Wertz, G., Yee, L. D., Goldstein, A. H., Carbone, S., Brito, J., Artaxo, P., de Gouw, J. A., Koss, A., Wisthaler, A., Mikoviny, T., Karl, T., Kaser, L., Jud, W., Hansel, A., Docherty, K. S., Alexander, M. L., Robinson, N. H., Coe, H., Allan, J. D., Canagaratna, M. R., Paulot, F., and Jimenez, J. L.: Characterization of a real-time tracer for isoprene epoxydiols-derived secondary organic aerosol (IEPOX-SOA) from aerosol mass spectrometer measurements, Atmos. Chem. Phys., 15, 11807-11833, 10.5194/acp-15-11807-2015, 2015.

McMeeking, G. R., Bart, M., Chazette, P., Haywood, J. M.,, Hopkins, J. R., McQuaid, J. B., Morgan, W. T., Raut, J.-C., Ryder, C. L., Savage, N., Turnbull, K., and Coe, H.: Airborne measurements of trace gases and aerosols over the London metropolitan region, Atmos. Chem. Phys., 12, 5163–5187, 2012.

Pieber, S. M., El Haddad, I., Slowik, J. G., Canagaratna, M. R., Jayne, J. T., Platt, S. M., Bozzetti, C., Daellenbach, K. R., Fröhlich, R., Vlachou, A., Klein, F., Dommen, J., Miljevic, B., Jimenez, J. L., Worsnop, D. R., Baltensperger, U., and Prévôt A. S. H.: Inorganic salt interference on CO2+ in Aerodyne AMS and ACSM organic aerosol composition studies, Environ. Sci. Tech., http://dx.doi.org/10.1021/acs.est.6b01035, 2016.

Xu, L., Guo, H., Boyd, C. M., Klein, M., Bougiatioti, A., Cerully, K. M., Hite, J. R., Isaacman-VanWertz, G., Kreisberg, N. M., Knote, C., Olson, K., Koss, A., Goldstein, A. H., Hering, S. V., de Gouw, J., Baumann, K., Lee, S.-H., Nenes, A., Weber, R. J., and Ng, N. L.: Effects of anthropogenic emissions on aerosol formation from isoprene and monoterpenes in the southeastern United States, Proceedings of the National Academy of Sciences, 112, 37-42, 10.1073/pnas.1417609112, 2015.

Xu, L., Williams, L. R., Young, D. E., Allan, J. D., Coe, H., Massoli, P., Fortner, E., Chhabra, P., Herndon, S., Brooks, W. A., Jayne, J. T., Worsnop, D. R., Aiken, A. C., Liu, S., Gorkowski, K., Dubey, M. K., Fleming, Z. L., Visser, S., Prévôt, A. S. H., and Ng, N. L.: Wintertime aerosol chemical composition, volatility, and spatial variability in the greater London area, Atmos. Chem. Phys., 16, 1139-1160, 10.5194/acp-16-1139-2016, 2016.

---

## Author Response (AR2)

**Author's response:**

We thank the Referees for the revision and comments which helped improving the quality of the manuscript. A point-by-point answer (in regular typeset) to the referees' remarks (in the *italic typeset*) follows, while changes to the manuscript are indicated in blue font. In the following page and lines references refer to the manuscript version submitted on 14th September 2016.

**Anonymous Referee #1**

*I recommend that the authors should add a short bit of text, either to the manuscript or the supplemental that addresses the reviewer 2's comment- that volatility and seasonal trends are linked.*

In the revised manuscript we replaced P22, L21-22 with the following discussion: In general, OOA factors with different seasonal behaviors can be characterized by different volatilities. However in this work the offline-AMS OOA separation is not driven by volatility, given the low correlation between $NO_3^-$ and our OOA factors (also reflected by the low $NO_3^-$-related SOA correlation with B-OOA and S-OOA, Table 2). Additionally, the partitioning of semi-volatile OA at low temperatures would lead to a less oxidized OOA fingerprint during winter than in summer; however, this was not the case. We observed a less oxidized OOA factor during summer, whose mass spectral fingerprint closely resembles that of SOA from biogenic precursors. Meanwhile similar to OOA from aging of biomass burning emissions, OOA during the cold season is more oxidized. This has been also reported in an urban environment in central Europe (Zurich) using an online-ACSM (Canonaco et al., 2015). Table 2 was moved below this section.

**Anonymous Referee #2**

**General Comments:**

*I thank the authors for taking time to revise the manuscript. The authors have addressed the comments adequately. However, I have two minor comments.*

*1. OM/OC ratio. While I agree with the authors that both Aiken and Canagaratna parameterizations are uncertain for this dataset, I want to point out that the OM/OC ratio would affect the recovery ratios determined by Eq. (6). Higher OM/OC ratio from Canagaratna parameterization would lead to lower recovery ratio and hence higher ambient concentration of factors.*

The recovery estimates are independent of the choice of Aiken or Canagaratna's OM:OC parameterizations. Indeed the recovery fitting equation (Eq. 6) explicitly contains the PMF factors OM:OC ratio. However the water-soluble PMF factor concentrations (Eq. 6) implicitly depend on the bulk OM:OC ratio used to determine the bulk WSOM concentration ($WSOM_i = WSOC_i OM/OC_i$) which was used as input for our PMF model. This leads to canceling corrections making the recovery estimates independent of the choice of the Aiken's or Canagaratna's OM:OC parameterizations.
This information was added in the revised SI.

*2. It is important to discuss why the same Rz is selected for both B-OOA and S-OOA (i.e., response to comment#20) and mention that the Rz of OOA factors warrants further investigation in the manuscript.*

The factor recoveries determined in this work enabled properly fitting the OC time series according to Eq. (6). The OC fitting residuals were unbiased within our uncertainty in different seasons (summer and winter) and at the different stations. Therefore there's no reason to consider statistically different recoveries for S-OOA and W-OOA.

We also fitted the factor recoveries according to Eq. (6) without any *a*-priori constrain from Daellenbach et al. (2016), and assuming different recoveries for S-OOA and B-OOA. The measured OC *vs.* fitted OC correlation was not statistically higher (95% confidence interval) than the correlation obtained when constraining the OOAs and BBOA factor recoveries according to Daellenbach et al. (2016). This suggests that the measured OC is equally well explained by the two fits.

The completely unconstrained fit returned a wide $R_{S\text{-OOA}}$ range (Fig. D10, only solutions associated to unbiased OC residuals and $R_k$s values comprised between 0 and 1 were retained). This occurs despite the considerable contribution of S-OOA, at all sites. This suggests that the least square algorithm fails to independently estimate the recoveries of factors and a priori constrains are needed to get unambiguous results. We have assumed $R_{S\text{-OOA}} = R_{B\text{-OOA}}$ based on the comparison between offline-AMS and online ACSM, although obtained at another site, especially that this assumption fits our knowledge of OOA water solubility and returned a mathematically equivalent OC reconstruction compared to the completely unconstrained model.

[Figure]

Fig. D10. $R_k$ probability density functions obtained by fitting Eq. (6) assuming $R_{S\text{-OOA}} \neq R_{B\text{-OOA}}$ and without *a*-priori $R_k$ information.

[revised manuscript text omitted]